# MEGA: Multilingual Evaluation of Generative AI

**Kabir Ahuja**[♡∗]   **Harshita Diddee**[◇∗]   **Rishav Hada** [†]   **Millicent Ochieng**[†]
**Krithika Ramesh** [♠∗]   **Prachi Jain**[†]   **Akshay Nambi**[†]   **Tanuja Ganu**[†]
**Sameer Segal**[†]   **Maxamed Axmed**[†]   **Kalika Bali**[†]   **Sunayana Sitaram**[†]
[♡]University of Washington   [◇]Carnegie Mellon University
[†]Microsoft Corporation   [♠]Johns Hopkins University
kahuja@cs.washington.edu, sunayana.sitaram@microsoft.com

## Abstract

Generative AI models have shown impressive performance on many Natural Language Processing tasks such as language understanding, reasoning, and language generation. An important question being asked by the AI community today is about the capabilities and limits of these models, and it is clear that evaluating generative AI is very challenging. Most studies on generative LLMs have been restricted to English and it is unclear how capable these models are at understanding and generating text in other languages. We present the first comprehensive benchmarking of generative LLMs - MEGA, which evaluates models on standard NLP benchmarks, covering 16 NLP datasets across 70 typologically diverse languages. We compare the performance of generative LLMs including Chat-GPT and GPT-4 to State of the Art (SOTA) non-autoregressive models on these tasks to determine how well generative models perform compared to the previous generation of LLMs. We present a thorough analysis of the performance of models across languages and tasks and discuss challenges in improving the performance of generative LLMs on low-resource languages. We create a framework for evaluating generative LLMs in the multilingual setting and provide directions for future progress in the field.

## 1 Introduction

Large Large Models (LLMs) such as ChatGPT and GPT-4 have created a lot of interest in the AI community and beyond, due to the step jump in their capabilities, such as maintaining context over conversations, fluency of generation, and reasoning. Many users have reported having tested these systems on languages other than English, with varying results, and recent demos of these models (Warren, 2023) have been shown in multiple (albeit high-resource) languages. Recently, the GPT-4 model

(OpenAI, 2023) was evaluated on the MMLU multiple choice questions benchmark by automatically translating it into 26 languages, and the results for some low-resource languages in the Latin script were found to be quite promising.

The multilingual capabilities of these models can be traced to their pre-training data, where even the predominantly English large-scale corpora contain hundreds of millions of non-English tokens (Blevins and Zettlemoyer, 2022). For GPT-3 unlabeled pre-training data has been documented to contain 119 languages (Brown et al., 2020), where roughly 93% of the tokens are in English[1]. Other LLMs like BLOOM (Scao et al., 2022) and PaLM (Chowdhery et al., 2022) have a better multilingual representation with 60% and 18% non-English data respectively for pre-training. While these models have been trained on multiple languages with varying distributions in the pre-training data, it is not clear how well they perform relative to each other across diverse tasks and languages due to a lack of comprehensive analysis across all models with the same experimental setup.

Recently, there has been a lot of interest in evaluating the different capabilities of LLMs, with comprehensive studies like HELM (Liang et al., 2022) that evaluate these models on a wide variety of capabilities. However, such studies are largely performed on English language data and there is a lack of such large-scale evaluation of LLMs for their multilingual capabilities. Given the current pace at which new language technologies are being developed that use LLMs, the importance of such an evaluation cannot be understated as the cases of inequalities in the performance of previous-generation models across languages have been well-documented (Blasi et al., 2022).

In our work, we present the first large-scale Multilingual Evaluation of Generative AI mod-

---

∗Work done when the author was at Microsoft.

[1]https://github.com/openai/gpt-3/blob/master/dataset_statistics/languages_by_word_count.csv

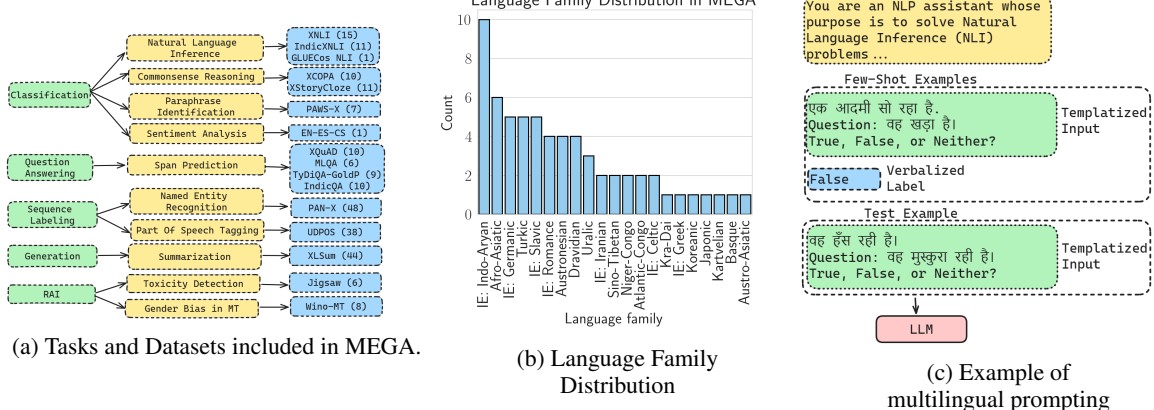

(a) Tasks and Datasets included in MEGA.

(b) Language Family Distribution

(c) Example of multilingual prompting

Figure 1: An overview of our benchmarking exercise: Multilingual Evaluation of Generative AI (MEGA). Numbers in parentheses in Figure 1a contain the number of languages supported in the dataset.

els (MEGA), spanning 16 different datasets, 70 topologically diverse languages, and four LLMs i.e. GPT-3.5 models `text-davinci-003` and `gpt-3.5-turbo`, GPT-4 (`gpt-4-32k`) and BLOOMZ (Muennighoff et al., 2022). We also compare these models with the models fine-tuned on these datasets like TULRv6 (Patra et al., 2022) and MuRIL (Khanuja et al., 2021), which are SoTA on different multilingual benchmarks.

Through our evaluation, we aim to answer three research questions. *(1)*, how well do LLMs fare on multilingual benchmarks compared to fine-tuned SOTA models? *(2)*, what languages do these models perform well in, and can we explain the trends in performance for these models across languages? *(3)*, what prompting strategies should be used for using LLMs for non-English languages?

Our study highlights that there is a significant disparity between the performance of LLMs in English vs non-English languages, especially low-resource languages with non-Latin scripts for which fine-tuned models perform significantly better. While GPT-4 bridges this gap to some extent, the discrepancy still exists. Further, we find that for these languages it is often difficult to do better than simply machine translating the input in a target language to English and then sending it to the LLM for prediction (*translate-test*). We also discuss how different prompt-design choices like prompt-tuning, use of explanations, and number of few-shot examples impact multilingual performance. Finally, we perform some initial analysis to the test the possibility of test data contamination in LLMs that we evaluate and discuss its implications on our findings. Our work provides a blueprint

for strategies that can be used for building systems using generative AI for multilingual users. We also release our code [2] for the community to scale up the multilingual evaluation of generative models.

## 2 MEGA

In this section, we discuss different components of our benchmarking exercise to measure the multilingual capabilities of LLMs. We start by discussing different NLP tasks and datasets that we evaluate these models on, along with their linguistic diversity. We provide an overview of the models we evaluate, baselines for comparison, and describe our evaluation scheme and prompting strategies.

### 2.1 Datasets and Languages

We broadly consider five families of NLP tasks in our experiments covering 16 different datasets:

**Classification Tasks.** Here, we further have four different sub-tasks, i) *Natural Language Inference* (classify if a hypothesis is entailed in the premise, contradicts it or neither), which includes XNLI (Conneau et al., 2018) , Indic-XNLI (Aggarwal et al., 2022) (version of XNLI translated to 11 Indian languages), and GLUECos NLI(Khanuja et al., 2020b) for English-Hindi code-mixed data; ii) *Commonsense Reasoning* datasets including causal commonsense reasoning benchmark XCOPA (Ponti et al., 2020) and XStoryCloze (Lin et al., 2022a), where the correct ending of a story with four sentences is to be predicted; iii) *Paraphrase Identification* task PAWS-X (Yang et al., 2019a), where given two sentences, the

---

[2] https://aka.ms/MEGA

model must predict if the two have the same meaning; iv) EN-ES-CS dataset for *Sentiment Analysis* on English-Spanish code-mixed tweets.

**Question Answering (QA).** For QA we consider *Span-Prediction* tasks, where the answer to a question is to be predicted within a piece of context provided. We evaluate on XQuAD (Artetxe et al., 2020), MLQA (Lewis et al., 2020), TyDiQA-GoldP (Clark et al., 2020), and IndicQA (Doddapaneni et al., 2022).

**Sequence Labeling.** This task involves classifying each token in a piece of text and we consider *Named Entity Recognition* dataset PAN-X (Pan et al., 2017) (also called WikiANN) and UDPOS (Nivre et al., 2018) for *Part of Speech Tagging*.

**Natural Language Generation (NLG).** For NLG we consider the multilingual *Abstractive Summarization* dataset XL-Sum.

**Responsible AI (RAI).** We consider the multilingual *Toxicity Prediction* dataset Jigsaw(Kivlichan et al., 2020), and Wino-MT to measure *Gender Bias* in MT systems.

All the datasets with the number of languages they include are listed in Figure 1a. These 16 datasets encompass a total of 70 languages covering 21 different language families, with Indo-Aryan and Afro-Asiatic languages in the majority (see Figure 1b). Note that for tasks with $> 30$ languages i.e. UDPOS, PAN-X, and XL-Sum, we run evaluations on the first 1000 examples of the test sets. For tasks where no public test sets are available (like XQUAD, TyDiQA-GoldP, and IndicQA), we evaluate on validation data. Refer to Appendix §A.1 for a detailed description of all the datasets.

## 2.2 Models

**OpenAI Models.** We conduct all benchmarking experiments on the GPT-3.5 models `text-davinci-003` (denoted as DV003 in the paper) and `gpt-3.5-turbo` (Ouyang et al., 2022) (GPT-3.5-Turbo) as well on the GPT-4 model `gpt-4-32k` (OpenAI, 2023). The `text-davinci-003` model has a maximum context size of 4096 tokens, while `gpt-3.5-turbo` and `gpt-4-32k` support context sizes of 16k and 32k respectively.

**Baselines.** We compare the performance of OpenAI models with two classes of baselines, i) *Prompt-Based baselines*, which like the OpenAI models are evaluated by prompting the model directly for solving a task, and ii) *Fine-tuned Base-*

*lines*, which are fine-tuned on task-specific training data. For the former we consider BLOOMZ (Muennighoff et al., 2022), a multi-task fine-tuned version of the BLOOM (Scao et al., 2022) model, which is a 176 billion parameter model trained on 46 natural languages and 13 programming languages. For fine-tuned baselines, we consider TULRv6 (Patra et al., 2022) (the current SoTA on XTREME benchmark), XLMR (Conneau et al., 2020), multilingual BERT (Devlin et al., 2019), and mT5 (Xue et al., 2021). For Indic-datasets we also compare with MuRIL(Khanuja et al., 2021), a multilingual BERT model trained on 16 Indic languages that obtains SOTA performance on many Indic benchmarks. All of these models (excluding mT5 for the XL-Sum and XCOPA), were fine-tuned with English data and then evaluated in a zero-cross-lingual fashion on other target languages.

## 2.3 Evaluation Methodology

LLMs exhibit two remarkable properties that make them effective at solving a variety of NLP tasks. The first is in-context learning (Brown et al., 2020), where the model learns to solve a task through the few input-output examples provided as part of the context without any weight updates. Secondly, the ability to follow instructions (Mishra et al., 2022; Wei et al., 2021; Ouyang et al., 2022) which is a property of instruction-tuned LLMs, where the models can be prompted to solve new-tasks based on the textual instructions provided in context.

We adopt these two techniques together to test the capabilities of LLMs to solve a variety of tasks in different languages. We define five main components to define the prompts: i) a **test example** $x_{\text{test}}$ for which the predictions are to be made; ii) $k$ **few-shot exemplars** $\{(x_i, y_i)\}_{i=1}^{k}$, that are used to provide in-context supervision to the model; iii) a **task instruction** $\mathcal{I}$ which describes the instruction in text for the task to LLM; iv) a **prompt template** $f_{\text{temp}}(x)$ which turns a dataset input example into a text format that can be used for prompting; and v) an **answer verbalizer** $f_{\text{verb}}(y)$ that maps the label $y$ to a textual representation. In our evaluation framework we often consider the instruction, template, and verbalizer as a single entity, and from now on will denote the template to encapsulate the three unless specified separately.

Given these components, the final prompt $f_{\text{prompt}}(x_{\text{test}}; \{(x_i, y_i)\}_{i=1}^{K}, \mathcal{I}, f_{\text{temp}}, f_{\text{verb}})$ or $f_{\text{prompt}}(x_{\text{test}})$ for short for a test input $x_{\text{test}}$ can

be defined as:

$$f_{\text{prompt}}(x_{\text{test}}) = \mathcal{I} \parallel_{i=1}^{K} \left\{ f_{\text{temp}}(x_i) \parallel f_{\text{verb}}(y_i) \right\}$$
$$\parallel f_{\text{temp}}(x_{\text{test}})$$

where $\parallel$ denotes the string concatenation operator. The prompt can then be provided as input to the LLM $P(.; \theta)$ to obtain the prediction $z_{\text{test}} = \arg\max_{z \in \mathcal{Z}} P(z | f_{\text{prompt}}(x_{\text{test}}); \theta)$, where $\mathcal{Z}$ is the space of possible answers, which in all of our experiments is taken to be the entirety of the language as modeled by the LLM. We approximate the $\arg\max$ by sampling from the probability distribution predicted by the LLM.

### 2.3.1 Multilingual Prompting Strategies

The choice of prompt significantly influences the performance of LLMs and these models have been shown to be brittle to simple prompting variations, such as the choice of prompt template and the training examples or even the ordering of examples (Zhao et al., 2021). For multilingual setups as highlighted in Lin et al. (2022a) and Shi et al. (2022), some additional variations to consider include, the choice of the language of the few-shot examples, the language of the prompt template, and the language of the test examples.

In this work, we evaluate models using three types of prompting strategies: **Monolingual Prompting**: In this setup, the $k$ randomly selected examples are of the same language as the test examples. **Zero-Shot Cross-Lingual**: Here, we evaluate generative models' zero-shot cross-lingual transfer ability during in-context learning. We use $k$-shot examples from a pivot language (always English in our experiments) which is different from the language of the test example. **Translate-Test**: In this setup also, the few-shot examples are sampled from English data. However, the test example itself is modified by translating it into English. We use Bing Translator to translate the test examples into English. We do not perform evaluations with Translate-Test prompting for QA and Sequence Labelling tasks where there is no trivial alignment between the labels in the translated text with native language text. To preserve costs, for GPT-4 we only run evaluations with the Monolingual prompting strategy except for a couple of datasets, which we explicitly discuss later in §3. Irrespective of the prompting strategy, we use the prompt templates written in English (see Appendix §A.7 for the impact of this choice).

**Prompt Tuning.** We use PromptSource (Bach et al., 2022) for a database of existing prompts to use for our experiments. In order to select the best prompt for a dataset (to appropriately measure the capabilities of these models), we evaluate the performance of available English templates on PromptSource on the English validation set and select the prompt that gives the best performance. This prompt template is then used to evaluate models on the test sets for all languages and prompt strategies. While it would be ideal to tune the prompts separately for each language, the scale of our experiments and the computational costs of these models make it prohibitive. We investigate the impact this choice has on our results in §4.1. We perform separate prompt-tuning for DV003 and GPT-3.5-Turbo models, and to keep the costs in check, we use the prompts obtained for the latter for GPT-4 as well. Final prompts selected are included in Appendix §A.4.

**Choice of Few-Shot Examples.** In all our experiments, we choose few-shot examples randomly from the training or validation set (depending on what's available) of a dataset. For most datasets, we use 8 few-shot examples, excluding tasks with longer contexts like QA and summarization tasks where we use $k = 4$.

## 3 Results and Analysis

In this section, we analyze the results of our benchmarking exercise across tasks and languages. Broadly, we cover the comparison between the effectivness of various prompting strategies §3.1 followed by the performance comparison of GPT-3.5 and GPT-4 models with appropriate baselines §3.2. We conclude with an examination of the factors that affects the performance of these models §3.3.

### 3.1 Comparing different prompting strategies

In Figure 2, we compare the performance of the three prompting strategies. We find that translate-test often improves the performance over the monolingual strategy, especially so in the case of DV003. We also find that for datasets, which include many low resource and non-latin script languages like IndicXNLI and XStoryCloze, the gains with translate-test are even more substantial for both the models. In Figure 3, we present the average (over different tasks) relative improvement by Translate-Test over Monolingual on GPT-3.5-Turbo for different languages and observe for languages like

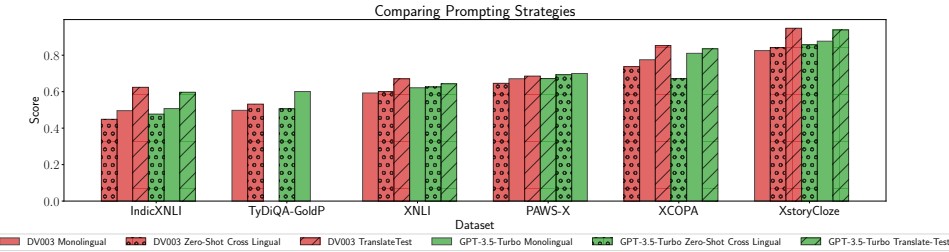

Figure 2: Comparing different prompting strategies discussed in §2.3.1 on DV003 and GPT-3.5-Turbo. The $y$-axis denotes the task-wise performance metric, e.g. Accuracy for XNLI and F1-Score for TyDiQA-GoldP. A list of metrics for all the tasks is provided in Table 1.

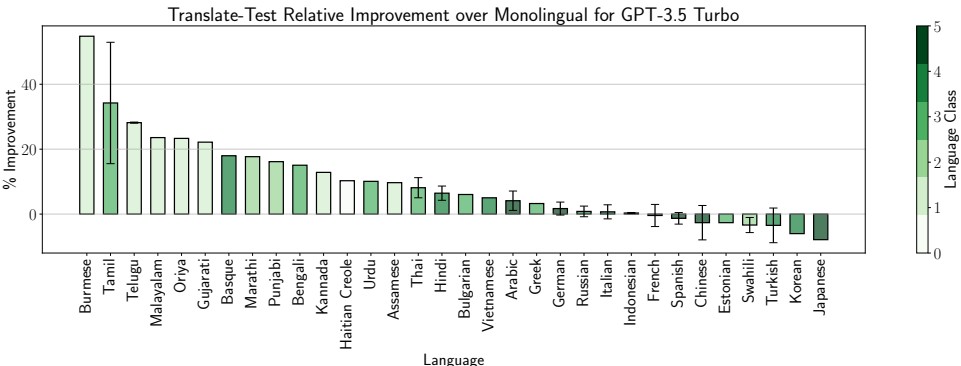

Figure 3: Relative percentage improvement over Monolingual prompting when using Translate-Test for GPT-3.5-Turbo. The bars are color-coded based on the class taxonomy provided in (Joshi et al., 2020)

Burmese, Tamil, and Telugu the relative improvement can be $> 30\%$! In general, we see that for low-resource languages, the translate-test results in substantial improvement in performance, while for high-resource languages the two perform similarly. While we do not evaluate GPT-4 Translate-Test exhaustively for all tasks, we do run the tests for XStoryCloze and XCOPA datasets. Based on these two, we observe that GPT-4's Monolingual prompting performance is often much more on-par with Translate-Test and many times even better. However, for low-resource languages we again see Translate-Test to perform much better. e.g., in XStoryCloze GPT-4's accuracy on Burmese is $77.6\%$ vs $93.2\%$ for Monolingual and Translate-Test respectively ( Figures 10b and 10d in Appendix).

Note that while Translate-Test substantially improves performance on low-resource languages, compared to the performance of these models in English, the gap even after Translate-Test is significantly high. For example, using translate-test with GPT-3.5-Turbo for Urdu in XNLI results in $54\%$ accuracy compared to $49.1\%$ for monolingual. However, this contrasts with the $76.2\%$ accuracy that the same model achieves in English.

Zero-Shot Cross-Lingual prompting for DV003

often performs on par with Monolingual but for GPT-3.5-Turbo, there is a drop in performance, especially so for tasks like XCOPA which have some extremely low resource languages: Quechua and Haitian Creole. For these languages, we observed that when provided few-shot examples in English, GPT-3.5-Turbo would often resort to predicting outputs like *"I'm sorry, but the premise is not in a language that I understand."*. However, by providing examples in the language, we are able to ground the model to these languages and we almost never observe such predictions in that case.

## 3.2 Comparing different models

The aggregated results comparing different models and prompting strategies are provided in Table 1 and Table 7 (for Indic Datasets). Excluding the commonsense reasoning tasks XCOPA and XStoryCloze, the OpenAI models generally lag behind the fine-tuned baseline TULRv6 for most tasks often by a significant margin and often are only slightly better than some of the smaller fine-tuned multilingual models i.e. mBERT and mT5-base. Between OpenAI models and BLOOMZ, the former models tend to outperform the latter (despite having a larger proportion of multilingual

| Model | Classification | | | | Question Answering | | | Sequence Labelling | | Summarization |
|---|---|---|---|---|---|---|---|---|---|---|
| | XNLI | PAWS-X | XCOPA | XStoryCloze | XQuAD | TyDiQA-GoldP | MLQA | UDPOS | PAN-X | XLSum |
| Metrics | Acc. | Acc. | Acc. | Acc. | F1 / EM | F1 / EM | F1 / EM | F1 | F1 | ROUGE-L |
| *Fine-tuned Baselines* | | | | | | | | | | |
| mBERT | 65.4 | 81.9 | 56.1 | × | 64.5 / 49.4 | 59.7 / 43.9 | 61.4 / 44.2 | 71.9 | 62.2 | × |
| mT5-Base | 75.4 | 86.4 | 49.9 | × | 67.0 / 49.0 | 57.2 / 41.2 | 64.6 / 45.0 | - | 55.7 | $\underline{28.1}^{\dagger}$ |
| XLM-R Large | 79.2 | 86.4 | 69.2 | × | 76.6 / 60.8 | 65.1 / 45.0 | 71.6 / 53.2 | 76.2 | 65.2 | × |
| TuLRv6 - XXL | $\mathbf{88.8}^{\dagger}$ | $\mathbf{93.2}^{\dagger}$ | $\mathbf{82.2}^{\dagger}$ | × | $\mathbf{86}$ / $\mathbf{72.9}^{\dagger}$ | $\mathbf{84.6}$ / $\mathbf{73.8}^{\dagger}$ | $\mathbf{81}$ / $\mathbf{63.9}^{\dagger}$ | $\mathbf{83.0}^{\dagger}$ | $\mathbf{84.7}^{\dagger}$ | × |
| *Prompt-Based Baselines* | | | | | | | | | | |
| BLOOMZ | 54.2 | $(82.2)^{\ddagger}$ | 60.4 | 76.2 | $(70.7 / 58.8)^{\ddagger}$ | $(75.2 / 63.2)^{\ddagger}$ | - | - | - | - |
| *Open AI Models* | | | | | | | | | | |
| `text-davinci-003` | 59.27 | 67.08 | 75.2 | 74.7 | 40.5 / 28.0 | 49.7 / 38.3 | 44.0 / 28.8 | - | - | - |
| `text-davinci-003 (TT)` | 67.0 | 68.5 | 83.8 | 94.8 | × | × | 54.9 / 34.6 | × | × | - |
| `gpt-3.5-turbo` | 62.1 | 70.0 | 79.1 | 87.7 | 60.4 / 38.2 | 60.1 / 38.4 | 56.1 / 32.8 | $60.2^{\ddagger}$ | 40.3 | 18.8 |
| `gpt-3.5-turbo (TT)` | 64.3 | 67.2 | 81.9 | 93.8 | × | × | 46.3 / 27.0 | × | × | 16.0* |
| `gpt-4-32k` | $75.4^{\ddagger}$ | 73.0 | $89.7^{\ddagger}$ | $96.5^{\ddagger}$ | 68.3 / 46.6 | 71.5 / 50.9 | $67.2 / 43.3^{\ddagger}$ | $66.6^{\ddagger}$ | $55.5^{\ddagger}$ | $19.7^{\ddagger}$ |

Table 1: Average performance across languages in each of the different datasets included in MEGA. TT suffix refers to the translate-test prompting strategy discussed in Section 2.3.1, without any suffix we refer to the monolingual strategy by default (except for XQuAD and IndicQA where it refers to cross-lingual setup). Numbers in **bold** with † symbol indicate best performing Fine-tuned model and the ones with ‡ refer to the best prompt-based generative model. The best overall numbers are underlined. For BLOOMZ the values in parenthesis indicate that the model was fine-tuned on the task during multi-task training. Missing values corresponding to the '×' symbol denote experiments that were not applicable and the ones with '-' were the ones deprioritized due to limited compute. `gpt-3.5-turbo (TT)` on XL-Sum was only evaluated on 29 languages which are supported by Bing Translator.

pre-training data), except for datasets like PAWS-X, XQUAD, and TyDiQA-GoldP, where BLOOMZ performs better. However, it must be noted that all these three datasets were present in the multi-task fine-tuning stage for BLOOMZ, especially for XQUAD and TyDiQA-GoldP for which the validation data that we use for evaluation is also likely to be included in the fine-tuning data[3].

Between the OpenAI models, generally DV003 and GPT-3.5-Turbo perform on par, with Translate-Test performance of DV003 being generally better than GPT-3.5-Turbo, and the other way around for Monolingual performance. However, we do observe a notable exception to this, which is for the QA tasks where GPT-3.5-Turbo performs substantially better than DV003, especially so for IndicQA. We attribute this to the fact that in order to fit the prompt in the 4096 context size for DV003, we had to resort to retrive-then prompt strategy and imperfect retrieval for low-resource languages leads to worse performance. Please check §A.5 of Appendix for more details on this. For GPT-4 on the other hand, we consistently observe substantial improvements, with it being *Pareto Optimal* (Choudhury and Deshpande, 2021) compared to the two GPT-3.5 models for all datasets with an exception of XL-Sum, where for some languages GPT-3.5-Turbo performs better. For the detailed

[3]Note that this can be a possibility for OpenAI models as well and we discuss this in more detail in §4.2.

results spanning all models, tasks, and languages, please refer to Appendix §A.8.

## 3.3 Factors Explaining Performance Trends

In this section, we try to understand what factors influence our observed trends in multilingual LLM capabilities. We begin by investigating the *Fertility* of the tokenizers used by different models, which is defined as the average number of sub-words produced per tokenized word (higher means worse quality), as that has been shown to critically impact the downstream task performance of pre-trained multilingual models (Rust et al., 2021). In Figure 4, we plot the tokenizer fertility of different models. We observe that the tokenizers for the OpenAI models are substantially worse for low-resource, non-latin script languages: where the fertility for languages like Malayalam and Tamil is so high ($\sim 10$) that the tokenizer essentially operates as a byte-level tokenizer for these languages. Note that this means that for low-resource languages, substantially larger number of tokens are needed to encode the inputs as well as for generation, which results in a significant additional API costs. Ahia et al. (2023) discusses how this phenomenon leads to large socio-economic disparities for speakers of underrepresented languages. We study if these discrepancies in the tokenizer's quality across languages have any effect on the performance. As can be seen in Figure 5, for six tasks we observe statis-

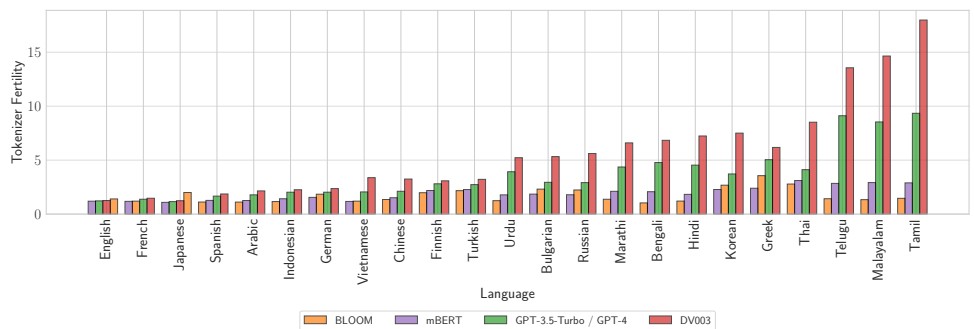

Figure 4: Tokenizer Fertility for OpenAI models, mBERT, and BLOOM for different languages

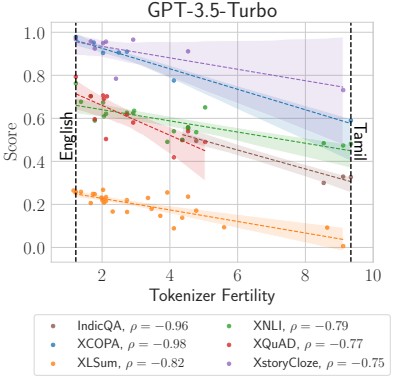

Figure 5: Correlation between the performance of GPT-3.5-Turbo with the tokenizer fertility. We report the curves for the cases where the person coefficient $|\rho| > 0.7$ with a p-value of 0.05. We have combined Indic-XNLI and XNLI for a better coverage of languages. Similar plots for GPT-4 can be found in Figure 7b of Appendix.

tically significant (negative) correlations between the tokenizer's fertility and dataset-specific performance i.e. the models obtain worse performance on languages for which the tokenizer is of poor quality, and vice-versa.

We also study the effect that the amount of data available for each language during pre-training (Wu and Dredze, 2020; Lauscher et al., 2020) has on the multilingual performance of these models. We measure the correlations between the language-wise number of tokens present in the pre-training data with language-wise performance on each dataset. While the exact language-wise pre-training data distribution for GPT-3.5 and GPT-4 models is not available, we use the GPT-3's language-wise pretraining distribution as a proxy. We observe that for four tasks (PAWS-X, XNLI, XCOPA, and XQuAD) statistically significant positive correlations between the pre-training data size and performance. Note that, the amount of pre-training data and tokenizer fertility are highly likely to be cor-

related with each other. However, we do see that using pre-training data we are able to explain some trends that are not explained by tokenizer fertility alone. For example, even though the OpenAI models have similar tokenizer fertilities for both French and Japanese, these models perform much better in French than they do for Japanese (72.1% accuracy vs 67% accuracy for GPT-3.5-Turbo) for PAWS-X. However, when we take into consideration the amount of pre-training data for these languages: roughly 3.5 B French tokens in the pre-training data versus 214M for Japanese, we can partially explain this discrepancy.

However, we must note that these two factors correlate well with only a subset of the tasks and what we are measuring is the correlation which might not imply causation. Investigating different factors that together more holistically explain multilingual capabilities is an important direction that we leave for future work. Please check Appendix §A.6 for detailed results from this section.

## 4 Challenges in Multilingual Evaluation

In this section, we examine some of the challenges and consequent limitations of a large-scale multilingual evaluation like ours.

### 4.1 A Kaleidoscope of Choices.

There are various moving parts when evaluating LLMs using prompting-based approaches, including the choice of prompt templates, instructions, and few-shot examples (Liu et al., 2022; Lu et al., 2022; Zhao et al., 2021), different prompting strategies (Wei et al., 2023; Nye et al., 2021; Ye and Durrett, 2022a), using external tools (Schick et al., 2023), the language of prompts (Shi et al., 2022; Lin et al., 2022a), as well as different decoding specific hyper-parameters (Shih et al., 2023), which can have varying degrees of impact on the performance, sometimes in unexpected ways. Holisti-

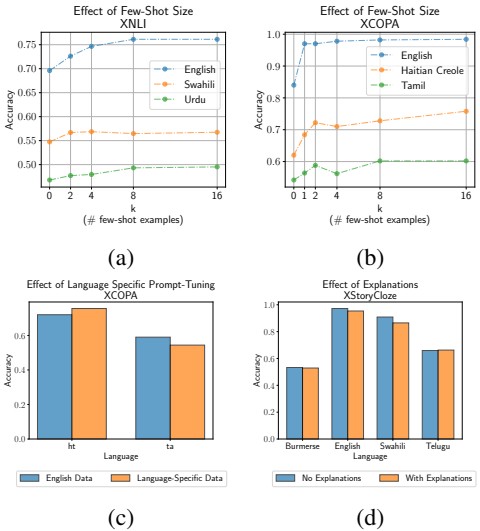

(a)  (b)

(c)  (d)

Figure 6: Analysing the effect on GPT-3.5-Turbo's performance given different evaluation factors. To obtain explanations we use Super-Natural Instructions (Wang et al., 2022).

cally exploring these choices for all the datasets and languages can quickly get out of hand, especially given the excessive computational cost of running these models. In order to understand the sensitivity of our observations to the choices we make in §3, we re-evaluate our setups on a subset of datasets and languages for a varying set of parameters. Our findings are summarized in Figure 6, where we see that having a large few-shot size generally helps improve performance, however, the performance is often stable beyond $k = 8$. Further, language-specific fine-tuning can help improve the performance like we see for Haitian Creole in XCOPA, but for Tamil we actually observe the accuracy to go down which might be attributed to the small size of the validation set (100 in the case of XCOPA). Finally, on XStoryCloze dataset (also for XCOPA), we see using explanations to prompt the models have negligible impact on the performance. Overall, these experiments indicate that the existing prompting approaches might not be sufficient to address the performance gap that exists for non-English languages (especially mid-to-low resource languages) and there is an imminent need to propose new methods as well as improve the representation of different languages in these model's pre-training (and instruction-tuning) data.

## 4.2 Test data contamination

Given the massive amount of online data that LLMs are trained with, it is critical to factor in the possibility of contamination of test datasets (Sainz et al.,

2023). Accordingly, we attempt to verify if the performances we observed are in fact, representative of the capabilities of these models or merely a result of memorization. Given the lack of transparency in the training distribution of recent models like GPT-4, we perform some preliminary investigations against this phenomenon. Specifically, we consider three factors: i) LLM's knowledge of the dataset, ii) availability of test datasets on the internet, and iii) dataset release date.

To measure the LLM's (we do this for GPT-4) memory of the dataset, we prompt it to fill the dataset cards for each of MEGA's datasets (denoted as Card Fill). This involves filling templatic information like the task's supported languages, input-output structure and description. If the model fills a dataset card correctly (Full), we note this as suspicion of contamination. If it fills the card partially correct (Partial) i.e. detecting either the correct task structure or correct set of languages, we mark it as partial evidence, and if it succeeds in neither, we mark it as no suspicion (None). For test dataset availability, we check if the test dataset can be accessed online directly without downloading either as part of the official release from the authors or via other sources such as Hugging Face dataset viewer (Data Acc. w/o Down.). For release date, we check if the dataset was made public after the cut-off date of September 2021.

The overall results from this analysis are provided in Table 2. We see that for a majority of datasets, GPT-4 can fill in the dataset card correctly; On the more recent datasets like XLSum and XStoryCloze it is only partially successful, while on Jigsaw and code-mixing datasets it fails to correctly fill the cards. Note that except XStoryCloze, Jigsaw and the Code-mixing datasets, evaluation sets for all other datasets are directly accessible online. Collectively, this connotes that for tasks like XStoryCloze and IndicQA there is a weak suspicion against contamination. While all other tasks are highly likely contaminated (except Jigsaw, and Code-Mixed datasets).

**Implications.** Our analysis implies a notable chance of the test data appearing in the training datasets of these LLMs. The contamination of test datasets is a serious problem for works centered around LLM evaluation (including ours), as they might lead to an overestimation of the capabilities of these models. However, we would like to highlight that despite the possibility of contamina-

| Dataset | Card Fill | Data Acc. w/o Down. | Release Date |
|---|---|---|---|
| XNLI | Full | Yes | September 2019 |
| Indic-XNLI | Full | Yes | April 2022 |
| PAWS-X | Full | Yes | August 2019 |
| XCOPA | Partial | Yes | April 2020 |
| XStoryCloze | Partial | No | May 2023 |
| XQuAD | Full | Yes | October 2019 |
| MLQA | Full | Yes | October 2019 |
| TyDiQA-GoldP | Full | Yes | February 2020 |
| IndicQA | Partial | Yes | September 2022 |
| PAN-X | Full | Yes | July 2017 |
| UDPOS | Full | Yes | March 2020 |
| XLSum | Partial | Yes | June 2021 |
| Jigsaw | None | No | February 2020 |
| GLUECos NLI | None | No | June 2020 |
| EN-ES-CS | None | No | May 2016 |

Table 2: Contamination analysis for the datasets that we consider in MEGA. We use red color when there is a strong suspicion of contamination based on these three metrics, green for no suspicion, and yellow for partial.

tion, LLMs still vastly underperform on (especially low-resource) non-English languages . These observations about data contamination indicate that the disparity in performance between English and non-English languages might be even greater than what we observe in our work.

## 5  Related Work

**Evaluation of LLMs.** A growing interest in the evaluation of LLMs has harbingered several efforts towards the holistic evaluation of their capabilities. While work like BIG-bench Srivastava et al. (2023) cover a diverse range of tasks, the non-English tasks are mostly translation-oriented which limit the more general task based inferences that for such an evaluation. Similarly, Liang et al. (2022) propose a taxonomy of scenarios and metrics in Holistic Evaluation of Language Models (HELM) to define the space of LLM evaluation, and evaluate 30 language models on 42 scenarios and 7 metrics. However, all the scenarios are focused on datasets in standard English or its dialects.

**Multilingual Benchmarks and Evaluation.** Benchmarks for multilingual evaluation, such as XTREME (Hu et al., 2020), XTREME-R (Ruder et al., 2021) and XGLUE (Liang et al., 2020) have been proposed to measure cross-lingual transfer in pre-trained language models. Following their popularity, there has been the development of benchmarks covering specific language families, such as IndicXTREME (Doddapaneni et al., 2022) for Indian languages, Adelani et al. (2022) for African Languages, and Wilie et al. (2020) for Indonesian languages, as well. The evaluations on these bench-

marks have mainly focused on pre-train then fine-tune kinds of setups. Particularly for prompting style evaluation, Bang et al. (2023) evaluates the multilingual capabilities of ChatGPT and shows that it fails to generalize to low-resource languages with non-latin scripts. However, multilingual evaluation is performed only on a few tasks, and a subset of 50-100 examples are used for testing the model. Hendy et al. (2023) evaluate the translation abilities of GPT-3.5 models and find that these models, while perform well in translating high-resource languages, their capabilities for low-resource languages are limited. Concurrent work BUFFET (Asai et al., 2023) and Lai et al. (2023) also perform multilingual benchmarking of large language models, however, they evaluate the performance of ChatGPT and BLOOMZ in their work while our evaluation also spans GPT-4.

**Multilingual Prompting:** While most work on prompting or in-context learning in LLMs focuses on English data, recently, there has been some interest in prompting them with non-English data. Zhao and Schütze (2021), for instance, use discrete and soft prompting techniques to evaluate XLM-RoBERTa and show that prompting can be more effective compared to fine-tuning when the amount of labeled data is limited. Lin et al. (2022a) show that English prompts perform better than prompts written in the target language (both hand-written and translated). Finally, (Shi et al., 2022) show chain-of-thought (CoT) prompting results leads to striking multilingual reasoning capabilities in LLMs, even in under-represented languages especially when prompted when English CoT.

## 6  Conclusion

In this work, we conduct an evaluation across different prompting strategies, models, tasks, and languages to investigate the multilingual capabilities of LLMs. We also investigate underlying properties like tokenizer quality and size of pretraining data to explain the trends in performance that we observe. Our investigation shows the consistent performance gap between high-resource, Latin script, and under-resourced languages in addition to highlighting the efficacy, yet limited sufficiency of methods like translate-test prompting. Through our evaluation, we present evidence of the need to prioritize automatic benchmarking and human evaluation across as many languages as possible. We hope that this work spurs research in meeting this goal.

## Limitations

Although we compare the evaluation results of GPT-3.5 and GPT-4 with BLOOMZ and SOTA models, we could not evaluate other closed models such as PaLM, which also contains training data in many languages. A limitation of our study is that we do not evaluate on all the multilingual datasets that are available, and we plan to scale up our evaluation in future versions of the study with the help of the research community. Even if we do evaluate all available multilingual datasets, they do not cover many typologically diverse and under-resourced languages, which is a fundamental limitation of trying to scale up multilingual evaluation today. For example, there is very little representation from African languages, Indigenous languages of the Americas etc. in any of the evaluation benchmarks available today. Finally, we restrict ourselves to the performance metrics and to some extent gender bias dimension of evaluation for this study - however, we plan to include evaluation of calibration, toxicity, bias, robustness, etc. in future work.

## Acknowledgments

The authors would like to thank Barun Patra and Vishrav Chaudhary for their help with TULR evaluation results. We also thank the anonymous reviewers for their helpful feedback, which helped us improve the quality of our paper.

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

# A Appendix

## A.1 Tasks and Datasets

In our experiments, we consider 16 tasks spanning the following task types - classification, sequence to sequence labeling and generation. Below we review the experimental setups and datasets used for benchmarking for these two tasks. A list of all the datasets with the languages covered by them can be found in Table 3.

### A.1.1 Classification

These tasks involve classifying a single sentence or a group of sentences into a finite number of discrete labels. For each dataset, we measure the performance of different models in terms of classification accuracy. For prompt-based models in particular, since we add no constraint on the output space of the LLM we compute the exact match between the generated output and a verbalized label to determine if the example was classified correctly. We run experiments for all the prompting strategies that we discussed in the previous sections for each dataset. The details of each dataset that we use for benchmarking are given below:

| Dataset | Task | Languages |
|---|---|---|
| XNLI | Natural Language Inference | 15 |
| Indic-XNLI | Natural Language Inference | 11 |
| GLUECoS | Natural Language Inference | 2 |
| PAWS-X | Paraphrase Identification | 7 |
| XCOPA | Commonsense Reasoning | 10 |
| XStoryCloze | Commonsense Reasoning | 11 |
| TyDiQA-GoldP | Question Answering | 9 |
| MLQA | Question Answering | 6 |
| XQuAD | Question Answering | 11 |
| IndicQA | Question Answering | 10 |
| UDPOS | Part of Speech Tagging | 38 |
| PANX | NER | 48 |
| WinoMT | Gender Bias | 8 |
| GLUECoS | Sentiment Analysis | 2 |
| Jigsaw | Toxicity Classification | 6 |
| XLSum | Summarization | 44 |

Table 3: Datasets and Language coverage of the datasets that MEGA presents evaluation for.

**1. Natural Language Inference**: XNLI (Conneau et al., 2018) is a dataset for cross-lingual Natural Language Inference, which consists of professional translations of the MNLI (Wang et al., 2018) corpus into 14 languages. We also consider IndicXNLI (Aggarwal et al., 2022) that translates the XNLI dataset into 11 Indic languages by using Machine Translation, followed by validation by native speakers.

**2. Paraphrase Identification**: PAWS-X (Yang et al., 2019b) is a paraphrase identification dataset professionally translated from the PAWS (Zhang et al., 2019) dataset into six typologically diverse languages.

**3. Commonsense Reasoning**: XCOPA (Ponti et al., 2020) is a commonsense reasoning dataset, which is a translation of the COPA (Roemmele et al., 2011) dataset into 11 typologically diverse languages, including very low-resource languages such as Eastern Apurímac Quechua and Haitian Creole.

XStoryCloze (Lin et al., 2022b) is created by translating the English StoryCloze (Mostafazadeh et al., 2017) dataset using professional translators into 10 typologically diverse languages.

### A.1.2 Question Answering

We focus on Span Prediction type of Question Answering (QA) tasks in our experiments, where given a context and a question the task is to predict the answer within the context. One major challenge that we come across for multilingual evaluation of QA tasks is that for many languages we often cannot fit the context and question pairs for the few-shot and text examples in the maximum context size of 4096 for the DV003 model. This is mainly attributed to the poor performance of GPT's tokenizer on many non-latin script languages which results in over-tokenizing the words in these languages.

To overcome this issue we follow two steps. First, for the few-shot examples we only provide the line within the paragraph containing the answer as the context. Second, for the test example, we index the chunks of the context using the embeddings from the `text-embedding-ada-002` model. Given the question, the closest chunk in the full context is retrieved and used in the prompt for the test example. We use a maximum chunk size of 100 in our experiments and use the implementation for retrieval provided in the **LangChain**[4] library. By doing this, we minimize the space taken by the context tokens in our prompt.

Note that, for newer GPT models i.e. GPT-3.5-Turbo and GPT-4 which support longer context lengths, we do not use this retrieval strategy for QA tasks and prompt the models to obtain the answers directly. For each task, we calculate the Exact Match and F1 score as defined in Rajpurkar

---

[4] https://github.com/hwchase17/langchain

et al. (2016a). For our experiments we consider the following four tasks:

**1. TyDiQA** (Clark et al., 2020) is a QA dataset covering 11 typologically diverse languages. The task consists of two sub-tasks - passage selection and minimum answer span (Gold-P). For our experiments, we consider the Gold-P task and evaluate Monolingual and Zero-Shot Cross-Lingual prompting strategies. Since the labels do not directly transfer one-to-one across translation for QA tasks as they do for classification and require the use of alignment algorithms, we skip translate-test prompting for this task.

**2. MLQA** (Lewis et al., 2020) is an extractive QA dataset translated into 7 languages by professional translators. The task has two variants, the first where the question, context, and answer are all in the same language; and the second, where the question is in a different language than the context and answer. We consider the former variant of the task in our experiments. For MLQA, translate-test splits are also available, where each language's test data has been translated into English with answers aligned using the attention scores. There is no training data available for MLQA, and we use SQuAD's Rajpurkar et al. (2016a) training data for selecting few-shot examples in English and validation data for MLQA in other languages to get their few-shot examples. This way, we are able to evaluate for all three prompting setups.

**3. XQuAD** (Artetxe et al., 2020) consists of professional translations of a subset of the SQuaD dataset (Rajpurkar et al., 2016b) into 10 languages. XQuAD only has validation datasets available publicly, hence we evaluate the models on them. Like MLQA we use English SQuAD data for few-shot examples and since we cannot use validation data in other languages for few-shot, we only evaluate for zero-shot cross-lingual setup for this task.

**4. IndicQA** (Doddapaneni et al., 2022) is a manually curated cloze-style reading comprehension dataset that can be used for evaluating question-answering models in 11 Indic languages. The context paragraphs are chosen from Wikipedia articles whose topics are closely related to Indic culture, history,etc. The publicly available test set has about 2000 sentences that we carry out our evaluation on.

## A.2    Sequences Labeling

In the sequence labeling task, a sequence of tokens (such as words) to be labeled are provided to the system.

### A.2.1    Part of Speech Tagging

UDPOS (Zeman et al., 2020) is a dataset for Part of Speech Tagging taken from the Universal Dependencies 2.5 from the XTREME (Hu et al., 2020) benchmark. We benchmark a subset of the languages available in UDPOS.

### A.2.2    Named Entity Recognition

PANX (Pan et al., 2017) or WikiANN is a Named Entity Recognition dataset consisting of Wikipedia sentences tagged with Person, Organization and Location.

For both tasks we use the linguistic structure prompting approach of Blevins et al. (2022) to define the prompts. The exact prompts used can be found in §A.4. Given the nature of both tasks, which would involve token alignment across the translation, we do not evaluate the translate-test prompting strategies for these setups. Also, since both tasks involve $> 30$ languages, to make the best use of the compute resources we only evaluate GPT-3.5-Turbo in a monolingual setup for these two tasks. Finally, we evaluate the first 1000 examples for each language for these datasets given the large number of languages. We have recomputed all baselines with this specification as well.

## A.3    Generation

### A.3.1    Summarization

The XLSum (Hasan et al., 2021a) dataset contains article-summary pairs across 44 typologically diverse languages, ranging from high to very low-resource.

For a similar reason as the tagging datasets, we only evaluate on first 1000 examples of the test sets in different languages and recompute the baselines on the same testset using the weights of the XL-SUM pretrained model, opensourced by the authors (Hasan et al., 2021b).

### A.3.2    Code-switching datasets

All the datasets we consider so far are monolingual, however, a majority of the world's population speaks more than one language, leading to language contact phenomena such as code-switching (Doğruöz et al., 2021; Sitaram et al., 2019). We include two code-switching datasets in MEGA to benchmark the performance of generative models.

GLUECoS-NLI (Khanuja et al., 2020a) is a code-mixed NLI dataset in Hindi-English, consist-

ing of Bollywood (Hindi) movie conversations as premises, with manually created hypotheses.

The EN-ES-CS Sentiment Analysis dataset (Vilares et al., 2016), part of the GLUECoS benchmark (Khanuja et al., 2020b) is a code-mixed dataset consisting of English-Spanish Tweets annotated with SentiStrength (Thelwall, 2017) scores.

### A.3.3 RAI datasets

We include two datasets that measure the Responsible AI (RAI) dimensions of fairness and toxicity - Jigsaw[5] for toxic comment classification and WinoMT for gender bias.

The Jigsaw dataset contains online comments sourced from Wikipedia. The training data, which is in English, contains labels pertaining to the toxicity of the comment and any relevant identity mentions contained in the comment. We use the test dataset, which contains these comments for 6 languages as illustrated in Table 3 for evaluation. The test dataset contains a binary label indicating whether or not the comment is toxic. Our objective is to assess the performance of these models across multiple languages and observe the disparity in this performance that could arise due to a number of factors, a prominent one being the source data that these models are trained on. Using English prompts from PromptSource for the original monolingual Jigsaw task, we task the model with classifying a comment as toxic or non-toxic. We perform crosslingual few-shot prompting and translate-test experiments for the test sets of all 6 languages, and report the results excluding content violations in Table 21.

The WinoMT dataset (Stanovsky et al., 2019) is created by concatenating the WinoGender (Rudinger et al., 2018) and WinoBias (Zhao et al., 2018) datasets. WinoMT dataset consists of 3888 English sentences with equal distribution of Male and Female genders. It is also equally balanced between stereotypical and non-stereotypical gender role assignments. We follow the method as reported by (Stanovsky et al., 2019) in their paper. We perform zero-shot monolingual prompting of all sentences in the dataset to translate them in 8 target languages. Further using *fast_align* we map the English entity to its translation. Finally, we extract the target-side entity's using off the shelf tools for each target language. The extracted translated

---

[5]https://www.kaggle.com/competitions/jigsaw-multilingual-toxic-comment-classification/data

gender can be finally compared against the gold annotations for English.

### A.4 Prompts

#### A.4.1 XNLI, IndicXNLI, GLUECoS NLI

**Models** : GPT-3.5-Turbo, GPT-4

Task Instruction $\mathcal{I}$: You are an NLP assistant whose purpose is to solve Natural Language Inference (NLI) problems. NLI is the task of determining the inference relation between two (short, ordered) texts: entailment, contradiction, or neutral. Answer as concisely as possible in the same format as the examples below:

Template $f_{temp}$:
{premise}
Question: {hypothesis}
True, False, or Neither?

Verbalizer $f_{verb}$:
Entailment : True,
Contradiction: False,
Neutral: Neither

**Models** : DV003

Template $f_{temp}$:
{premise} Based on previous passage is it true that {hypothesis} ? Yes, No, or Maybe?

Verbalizer $f_{verb}$:
Entailment : Yes,
Contradiction: No,
Neutral: Maybe

#### A.4.2 PAWS-X

**Models** : GPT-3.5-Turbo, GPT-4

Task Instruction $\mathcal{I}$: You are an NLP assistant whose purpose is to perform Paraphrase Identification. The goal of Paraphrase Identification is to determine whether a pair of sentences have the same meaning. Answer as concisely as possible in the same format as the examples below:

Template $f_{temp}$:
{sentence1}
Question: {sentence2}
True or False?

**Models** : DV003

Template $f_{temp}$:
Sentence 1: {sentence1} Sentence 2: {sentence2} Question: Does Sentence 1 paraphrase Sentence 2 ? Yes or No?

Verbalizer $f_{verb}$:
*Positive*: Yes
*Negative*: No

### A.4.3 XCOPA

**Models** : GPT-3.5-Turbo, GPT-4

Task Instruction $\mathcal{I}$: You are an AI assistant whose purpose is to perform open-domain commonsense causal reasoning. You will be provided a premise and two alternatives, where the task is to select the alternative that more plausibly has a causal relation with the premise. Answer as concisely as possible in the same format as the examples below:

Template $f_{temp}$:
```
{ premise }
{% if question == "cause" %} This happened
because...
{% else %} As a consequence... {% endif %}
```
Help me pick the more plausible option: -
`{choice1}`-`{choice2}`

**Models** : DV003

Template $f_{temp}$:
```
{ premise }
{% if question == "cause" %} This happened
because...
{% else %} As a consequence... {% endif %}
```
Help me pick the more plausible option: - choice1:
`{choice1}`, choice2: `{choice2}`

Verbalizer $f_{verb}$:
choice1: `{choice1}`
choice2: `{choice2}`

### A.4.4 XQUAD, TyDiQA, MLQA

**Models** : GPT-3.5-Turbo, GPT-4

Task Instruction $\mathcal{I}$: You are an NLP assistant whose purpose is to solve reading comprehension problems. You will be provided questions on a set of passages and you will need to provide the answer as it appears in the passage. The answer should be in the same language as the question and the passage.

Template $f_{temp}$:
```
{context}
Q: {question}
```
Referring to the passage above, the correct answer to the given question is: `{answer}`

**Models** : DV003

Template $f_{temp}$:

```
{context}
Q: {question}
```
Referring to the passage above, the correct answer to the given question is: `{answer}`

### A.4.5 IndicQA

**Models** : GPT-3.5-Turbo, GPT-4

Task Instruction $\mathcal{I}$: You are an NLP assistant whose purpose is to solve reading comprehension problems. You will be provided questions on a set of passages and you will need to provide the answer as it appears in the passage. The answer should be in the same language as the question and the passage.

Template $f_{temp}$:
```
{context}
Q: {question}
```
Referring to the passage above, the correct answer to the given question is? If you can't find the answer, please respond "unanswerable". `{answer}`

**Models** : DV003

Template $f_{temp}$:
```
{context}
Q: {question}
```
Referring to the passage above, the correct answer to the given question is: `{answer}`

### A.4.6 XStoryCloze

**Models** : DV003, GPT-3.5-Turbo, GPT-4

Template $f_{temp}$:
```
{input_sentence_1}    {input_sentence_2}
{input_sentence_3} {input_sentence_4}
```
What is a possible continuation for the story given the following options ?
Option1:     `{sentence_quiz1}`    Option2:
`{sentence_quiz2}`

Verbalizer $f_{verb}$:
`{sentence_quiz1}`: Option1,
`{sentence_quiz2}`: Option2

### A.4.7 PANX

**Models** : GPT-3.5-Turbo, GPT-4

Task Instruction $\mathcal{I}$: You are an NLP assistant whose purpose is to perform Named Entity Recognition (NER). NER involves identifying and classifying named entities in a text into predefined categories such as person names, organizations, locations, and others. You will need to use the tags defined below: O means the word doesn't correspond to any entity. B-PER/I-PER means the word corresponds to the

beginning of/is inside a person entity. B-ORG/I-ORG means the word corresponds to the beginning of/is inside an organization entity. B-LOC/I-LOC means the word corresponds to the beginning of/is inside a location entity. Do not try to answer the question! Just tag each token in the sentence.

Template $f_{temp}$: `{token_1 token_2 ... token_n}`

Verbalizer $f_{verb}$:
```
{tag_1} {tag_2} ... {tag_n}:
{token_1}_{tag_1} {token_2}_{tag_2}
... {token_n}_{tag_n}
```

### A.4.8 UDPOS

**Models** : GPT-3.5-Turbo, GPT-4

Task Instruction $\mathcal{I}$: You are an NLP assistant whose purpose is to perform Part of Speech (PoS) Tagging. PoS tagging is the process of marking up a word in a text (corpus) as corresponding to a particular part of speech, based on both its definition and its context. You will need to use the tags defined below:

1. ADJ: adjective
2. ADP: adposition
3. ADV: adverb
4. AUX: auxiliary
5. CCONJ: coordinating-conjunction
6. DET: determiner
7. INTJ: interjection
8. NOUN: noun
9. NUM: numeral
10. PART: particle
11. PRON: pronoun
12. PROPN: proper-noun
13. PUNCT: punctuation
14. SCONJ: subordinating-conjunction
15. SYM: symbol
16. VERB: verb
17. X: other

Template $f_{temp}$: `{token_1 token_2 ... token_n}`

Verbalizer $f_{verb}$:
```
{tag_1} {tag_2} ... {tag_n}:
{token_1}_{tag_1} {token_2}_{tag_2} ...
{token_n}_{tag_n}
```

### A.4.9 GLUECoS Sentiment Analysis

**Models** : GPT-3.5-Turbo, GPT-4

Task Instruction $\mathcal{I}$: You are an NLP assistant whose purpose is to solve Sentiment Analysis problems. Sentiment Analysis is the task of determining whether the sentiment, opinion or emotion expressed in a textual data is: positive, negative, or neutral. Answer as concisely as possible in the same format as the examples below:

Template $f_{temp}$: Does the following sentence have a positive, negative or neutral sentiment? `{text}`

**Models** : DV003

Template $f_{temp}$: Does the following sentence have a positive, negative or neutral sentiment? `{text}`

### A.4.10 XLSum

**Models** : GPT-3.5-Turbo, GPT-4

Task Instruction $\mathcal{I}$: You are an NLP assistant whose purpose is to summarize any given article. You should summarize all important information concisely in the same language in which you have been provided the document. Following the examples provided below:

Template $f_{temp}$:
```
{document}
===
```
Write a summary of the text above :

### A.4.11 Jigsaw

**Models** : GPT-3.5-Turbo, GPT-4

Task Instruction $\mathcal{I}$: You are an NLP assistant whose purpose is to solve the Jigsaw Toxicity Prompt problem. Toxicity is defined as anything rude, disrespectful or otherwise likely to make someone leave a discussion. The objective is to identify toxic comments. Answer as concisely as possible in the same format as the examples below:

Template $f_{temp}$: `{text}` Should this online comment be removed for its toxicity? Yes or No?

**Models** : DV003

Template $f_{temp}$: `{text}` Someone posted this comment online. Do you find it toxic? Yes or No?

### A.4.12 WinoMT

**Models** : GPT-3.5-Turbo, GPT-4

Template $f_{temp}$: Translate the following English text to `{target_language}`: `{sentence}`

| Language | Retrieval Acc. |
|----------|----------------|
| en | 0.858 |
| ar | 0.492 |
| bn | 0.141 |
| fi | 0.756 |
| id | 0.680 |
| sw | 0.760 |
| ko | 0.453 |
| te | 0.056 |
| ru | 0.421 |

Table 4: Retrieval accuracy on TyDiQA dataset for chunk size = 100.

## A.5 Handling Long Contexts

As discussed in §2.2, the models we study have limited context lengths and for QA tasks in particular, fitting the entire prompt containing the few-shot examples is often not feasible for low-resource languages where the tokenizers of these models are found to over-tokenize the text (nearly resulting in byte level tokens). To overcome this issue we follow two steps. First, for the few-shot examples we only provide the line within the paragraph containing the answer as the context. Second, for the test example, we index the chunks of the context using the embeddings from the `text-embedding-ada-002` model. Given the question, the closest chunk in the full context is retrieved and used in the prompt for the test example. We use a maximum chunk size of 100 in our experiments and use the implementation for retrieval provided in the **LangChain**[6] library. By doing this, we minimize the space taken by the context tokens in our prompt. Note that, for newer GPT models i.e. GPT-3.5-Turbo and GPT-4 which support longer context lengths, hence we only use this retrieval strategy for DV003 on QA tasks.

We attribute the significantly worse performance of DV003 on IndicQA to imperfect retrieval in the case of DV003, while for GPT-3.5-Turbo we do not rely on retrieval due to the larger context size. We provide the retrieval accuracies for DV003 (i.e. if the retrieved chunk contains the answer) in Appendix Table 4 , where we clearly see for low-resource languages like Telugu, the accuracies can be as low as 5%. While beyond the scope of this work, alternate retrieval strategies like using better embeddings from multilingual models for retrieval can be explored to close this gap (Nambi et al., 2023).

---
[6] https://github.com/hwchase17/langchain

## A.6 Factors Explaining Multilingual Capabilities of LLMs

We provide correlation plots in Figures 7 (between performance and fertility) and 8 (between performance and pre-training size) for both GPT-3.5-Turbo and GPT-4. The exact values of the correlations for all tasks and the two models is provided in Table 5.

## A.7 Challenges in Multilingual Evaluation

**Effect of number of in-context examples $k$.** Our main experiments were conducted with $k = 8$ or $k = 4$, depending on the task. Here, we evaluate what effect different numbers of in-context examples have on XNLI and XCOPA for three languages in Figures 6a and 6b. We observe while the performance increases sharply while moving from 0 to 2-4 examples, it is fairly stable after $k \geq 8$, with the exception of Haitian Creole in XCOPA, where it continues to improve.

**Effect of language-specific prompt tuning.** As discussed in §2.3.1, we use English validation data for prompt selection in each dataset that we use for all languages. Here, we explore whether separately tuning the prompts for each language helps. For XNLI, we run this experiment on Urdu and Swahili, tuning over ten different prompt templates from Prompt-Source, but find that the same prompt that was tuned for English gets picked up for these two languages as well. For XCOPA however, different prompts are chosen when tuned on Haitian Creole and Tamil. This leads to an improvement in the test performance for Haitian Creole (from 72% to 75.6%, see Figure 6c). Interestingly for Tamil, we see the test performance actually drops slightly compared to the accuracy obtained with prompt selected on English data, which we conjecture might be due to the fact that the validation sets in XCOPA have only 100 examples that may not be sufficient for selecting optimal prompts.

**Effect of Explanations.** Ye and Durrett (2022b), showed for `text-davinci-002`, that prompting the model with explanations before the outputs (Explain-then-Predict) in the in-context examples can help improve few-shot performance substantially on English language datasets. Hence, here we evaluate if they help improve the multilingual performance of the GPT-3.5-Turbo model as well. We perform experiments on XStoryCloze and XCOPA datasets and use the explanations available in Super-

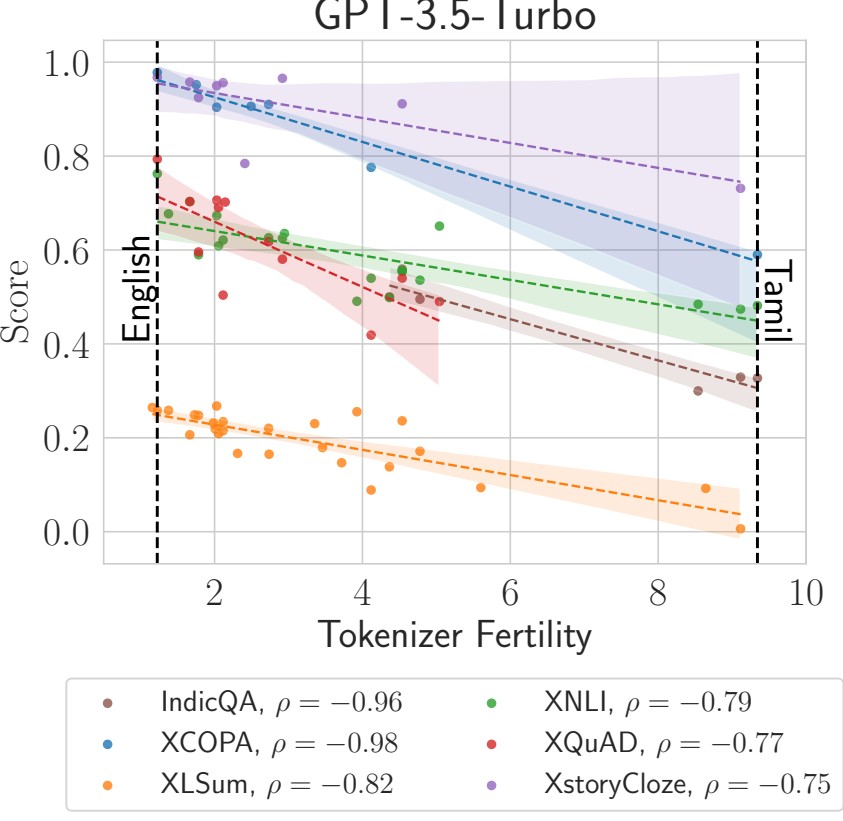

(a) Correlation between tokenizer fertility and performance for GPT-3.5-Turbo.

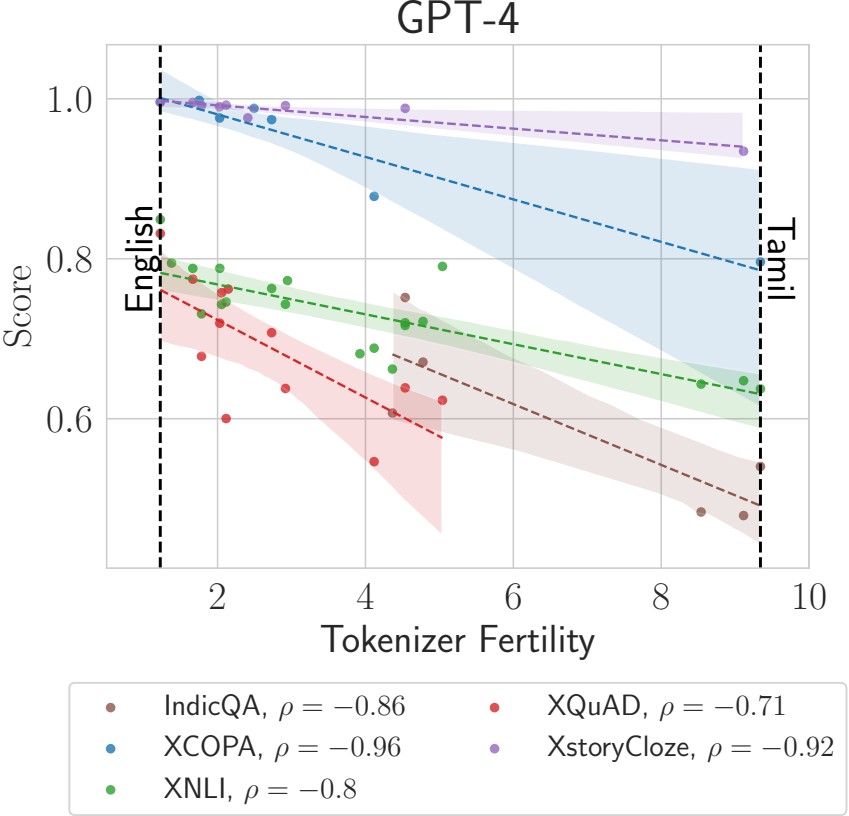

(b) Correlation between tokenizer fertility and performance for GPT-4

Figure 7: Correlation between the performance of GPT-3.5-Turbo and GPT-4 with the tokenizer fertility.

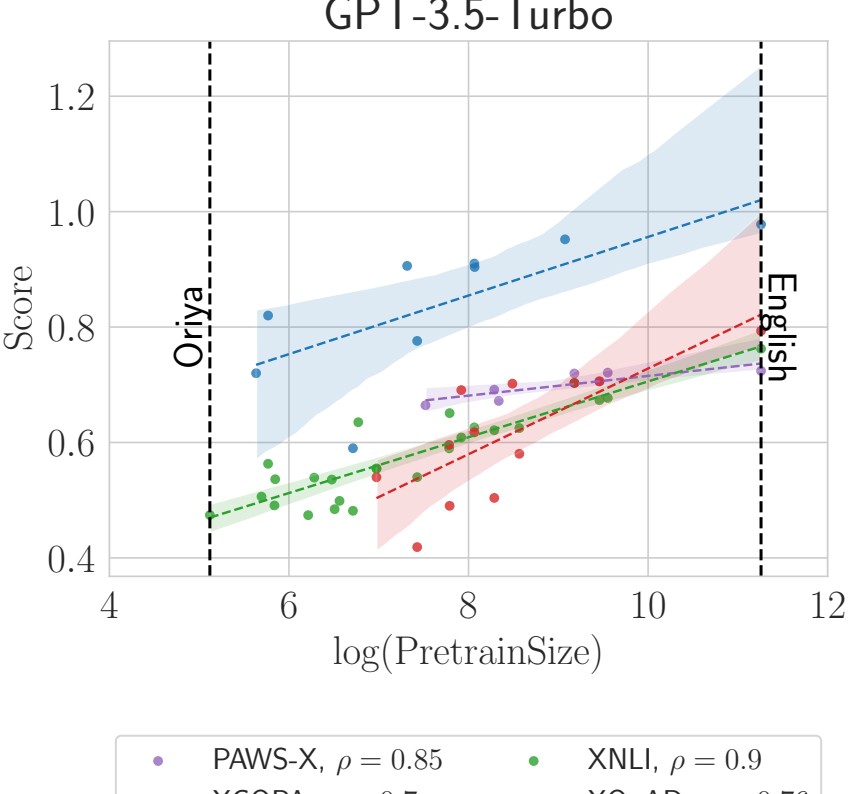

(a) Correlation between pre-training size and performance for GPT-3.5-Turbo.

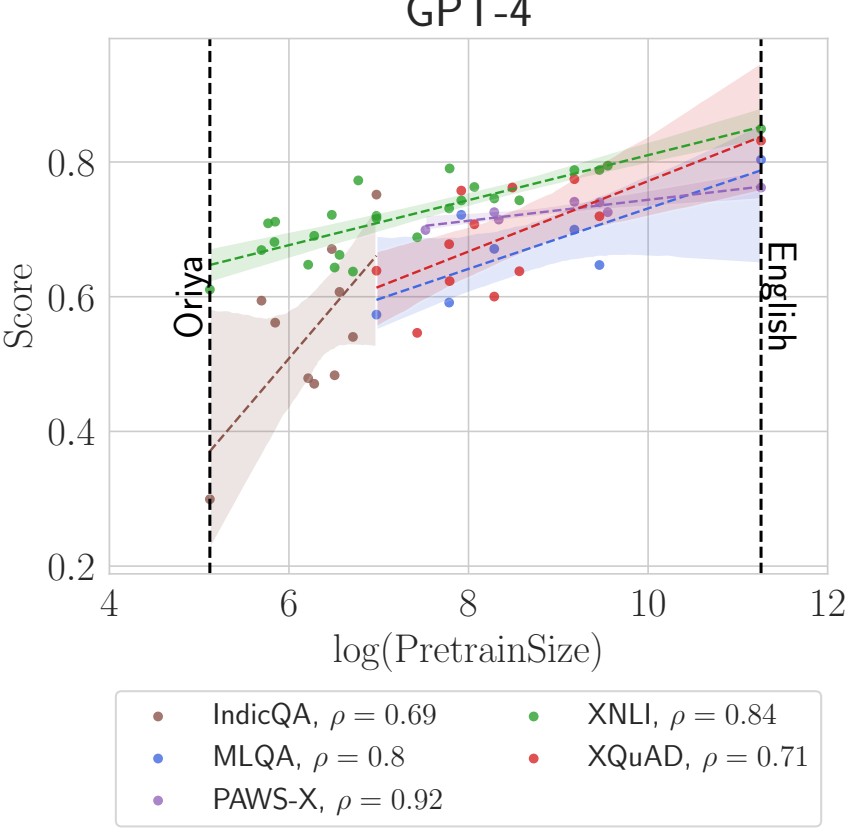

(b) Correlation between pre-training size and performance for GPT-4

Figure 8: Correlation between the performance of GPT-3.5-Turbo and GPT-4 with the pre-training size.

| | Tokenizer Fertility | | | | Pre-training Size | | | |
| | GPT-3.5-Turbo | | GPT-4 | | GPT-3.5-Turbo | | GPT-4 | |
| Task | $\rho$ | P-value | $\rho$ | P-value | $\rho$ | P-value | $\rho$ | P-value |
|---|---|---|---|---|---|---|---|---|
| XNLI+IndicXNLI | $-0.784$ | $6.9e-05$ | $-0.803$ | $3.4e-05$ | $0.893$ | $4.1e-09$ | $0.836$ | $3.5e-07$ |
| XCOPA | $-0.982$ | $7.9e-05$ | $-0.957$ | $0.00$ | $0.70$ | $0.035$ | $0.489$ | $0.181$ |
| XstoryCloze | $-0.745$ | $0.033$ | $-0.918$ | $0.001$ | $0.603$ | $0.064$ | $0.407$ | $0.242$ |
| PAWS-X | $-0.587$ | $0.219$ | $-0.61$ | $0.198$ | $0.85$ | $0.031$ | $0.94$ | $0.005$ |
| MLQA | $-0.451$ | $0.368$ | $-0.674$ | $0.141$ | $0.71$ | $0.085$ | $0.808$ | $0.051$ |
| TyDiQA-GoldP | $0.543$ | $0.163$ | $0.049$ | $0.907$ | $-0.464$ | $0.207$ | $-0.159$ | $0.682$ |
| XQuAD | $-0.865$ | $0.00$ | $-0.818$ | $0.002$ | $0.782$ | $0.004$ | $0.736$ | $0.009$ |
| IndicQA | $-0.960$ | $0.002$ | $-0.856$ | $0.029$ | $0.628$ | $0.051$ | $0.690$ | $0.027$ |
| WinoMT | $-0.36$ | $0.379$ | - | - | $0.249$ | $0.589$ | - | - |
| Jigsaw | $0.306$ | $0.554$ | - | - | $-0.674$ | $0.141$ | - | - |
| PAN-X | $-0.456$ | $0.003$ | $-0.442$ | $0.004$ | $0.41$ | $0.006$ | $0.326$ | $0.032$ |
| UDPOS | $-0.216$ | $0.198$ | $-0.304$ | $0.066$ | $0.29$ | $0.095$ | $0.359$ | $0.036$ |
| XLSum | $-0.821$ | $4.8e-07$ | $-0.578$ | $0.002$ | $0.448$ | $0.011$ | $0.49$ | $0.005$ |

Table 5: Pearson Correlation coefficient $\rho$ between performance and tokenizer fertility and performance and pre-training data size for different datasets and models. We also provide the p-values, to see which correlations are statistically significant.

NaturalInstructions (SNI)(Wang et al., 2022)[7]. All the explanations that we used were written in English. For XStoryCloze, the results are plotted in Figure 6d, and we observe that while there is a slight gain upon using explanations for Telugu, for all other languages the performance remains largely unchanged if not slightly worse. Interestingly, upon manual inspection of the model's prediction, we observe that the model often first translates the problem to English and then proceeds with the explanation, without having prompted to do so. We have similar observations for the XCOPA dataset as well, where adding explanations doesn't help improve performance and ends up hurting the performance by a slight margin (Figure 9)

**Effect of the language of the prompt templates.** While all our experiments were run using prompt templates written in English, we initially evaluated DV003 on Native-Language-Templates as well, which were obtained by translating English templates using Bing-Translator. As can be seen in Table 6, the performance is much worse when using templates in the native language compared to English. This is consistent with the results in Muennighoff et al. (2022) for BLOOMZ and Lin et al.

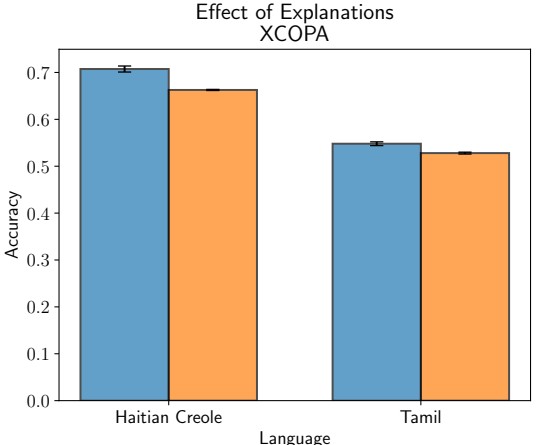

Figure 9: Effect of using explanations in XCOPA dataset. Blue bars mean no explanations in the prompt and orange bars correspond to prompting with explanations.

---

[7]At the time of writing this paper, XStoryCloze wasn't included in SNI, hence we use the few-shot examples and explanations available for StoryCloze dataset(Mostafazadeh et al., 2016), making the prompting setup Zero-Shot Cross-Lingual.

| Task | English-Template | Native-Language-Template |
|---|---|---|
| XNLI | **58.3** | 54.4 |
| Indic-XNLI | **49.6** | 38.7 |
| PAWS-X | **67.1** | 64.2 |
| XCOPA | **77.6** | 73.1 |

Table 6: Average performance on non-English languages with the Monolingual Prompting strategy using English-Template and Native-Language-Template prompts for the classification tasks for DV003.

| Model | IndicXNLI | IndicQA |
|---|---|---|
| MuRIL | **72.4** | 47.7 |
| text-davinci-003 | 49.6 | 8.45 |
| text-davinci-003 (TT) | 62.4 | × |
| gpt-3.5-turbo | 50.7 | 38.6 |
| gpt-3.5-turbo (TT) | 59.7 | × |
| gpt-4-32k | 66.8 | **55.0** |

Table 7: Comparing performance of text-davinci-003, gpt-3.5-turbo, and gpt-4-32k with fine-tuned baseline MuRIL on Indic datasets (Doddapaneni et al., 2022). For IndicXNLI we report Accuracy and F1-score for IndicQA.

(2022a) for XGLM, which also show better performance when using prompt templates in English.

## A.8 Detailed Results

The results for across all tasks, languages and models included in our benchmarking exercise can are provided in Figures 10 (for Classification tasks), 11 (for QA Tasks), 12 (for XLSum), 14 (for PAN-X), 13 (for UDPOS), 15 (for Jigsaw), and finally 16 (for Wino-MT). The results for the Indic Datasets and the two code-mixed datasets GLUECoS NLI and En-ES-CS are provided in Tables 7 8 respectively.

| Model | NLI En-Hi | Sentiment En-Es |
|---|---|---|
| mBERT | 63.1 | 69.31 |
| text-davinci-003 | 72.1 | **68.8** |
| gpt-3.5-turbo | **78.8** | 68.0 |

Table 8: Performance of GPT-3.5 models on code-mixing datasets from (Khanuja et al., 2020b). For both the tasks, the metric reported is accuracy.

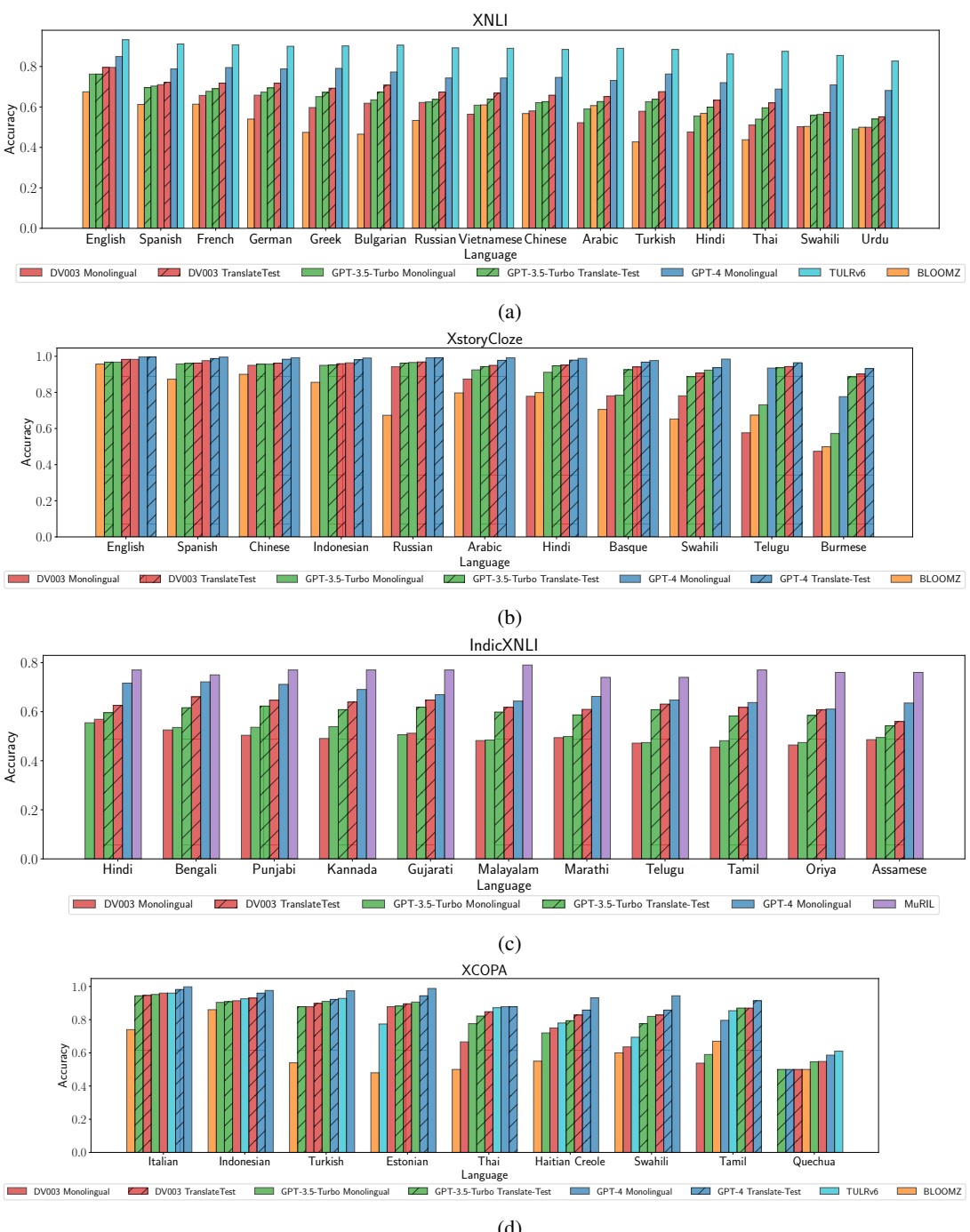

Figure 10: Comparing the language-wise performance of different models on the classification tasks.

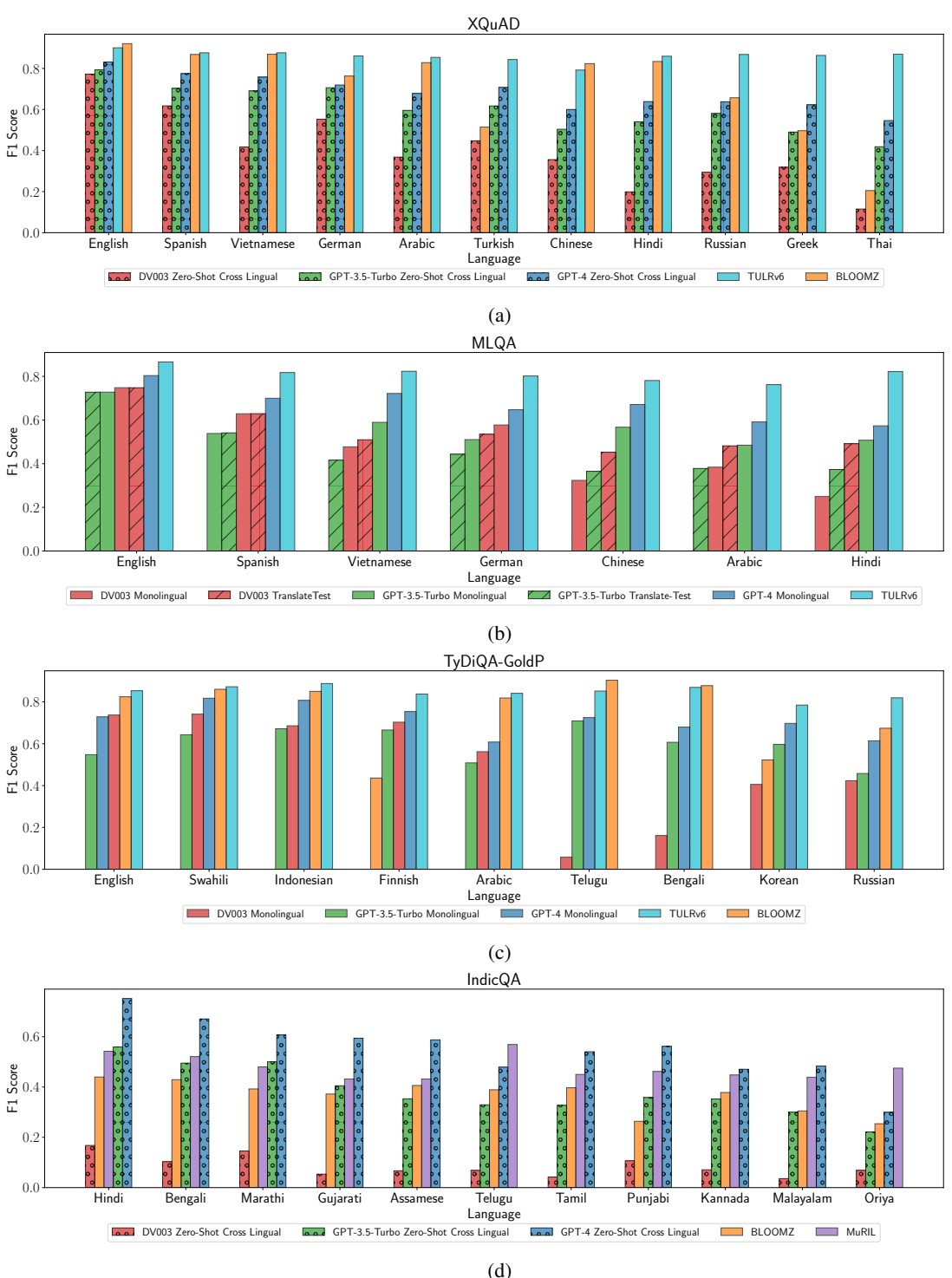

Figure 11: Comparing the language-wise performance of different models on the QA tasks.

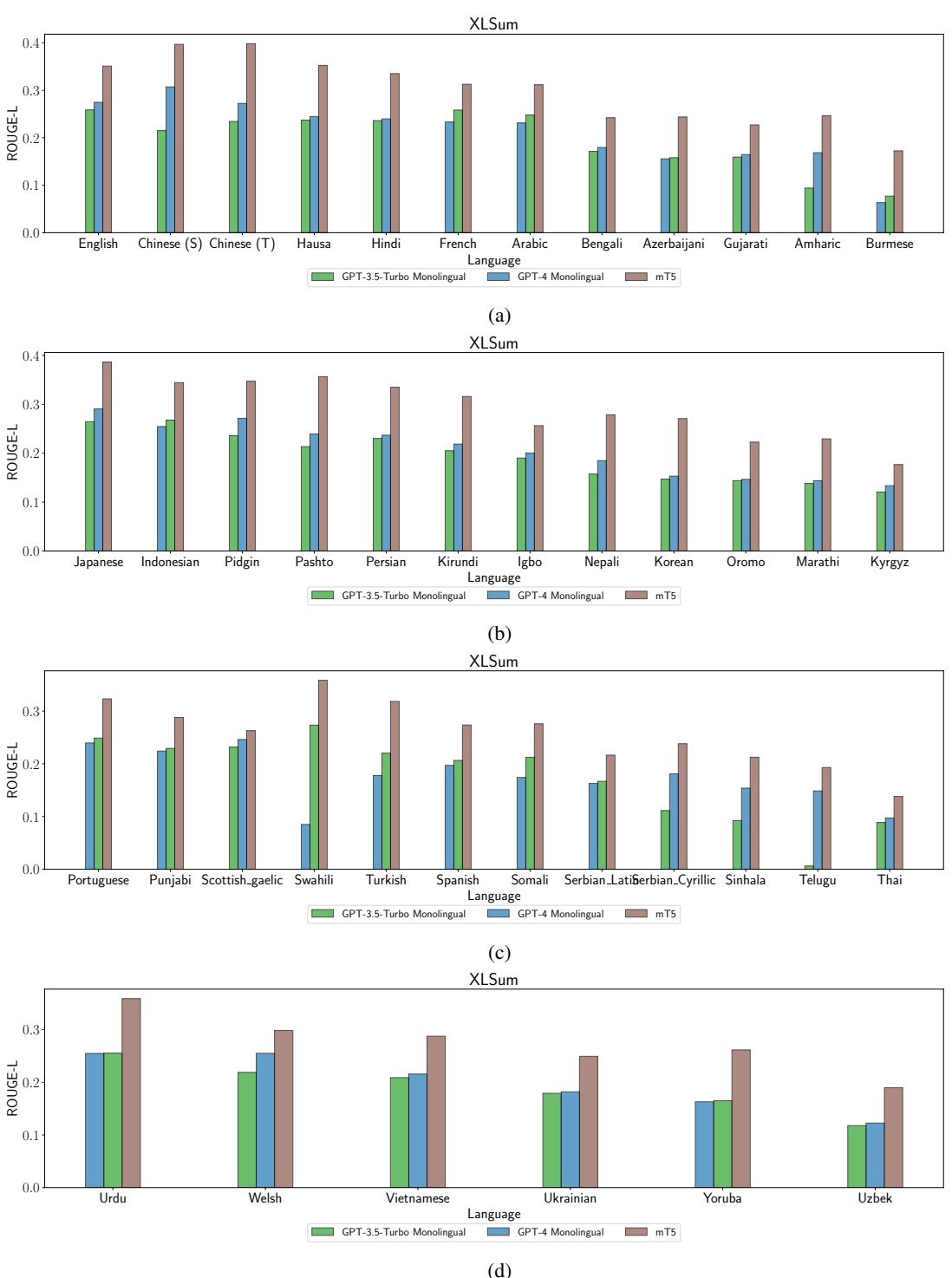

Figure 12: Comparing performance of different models on XLSUM.

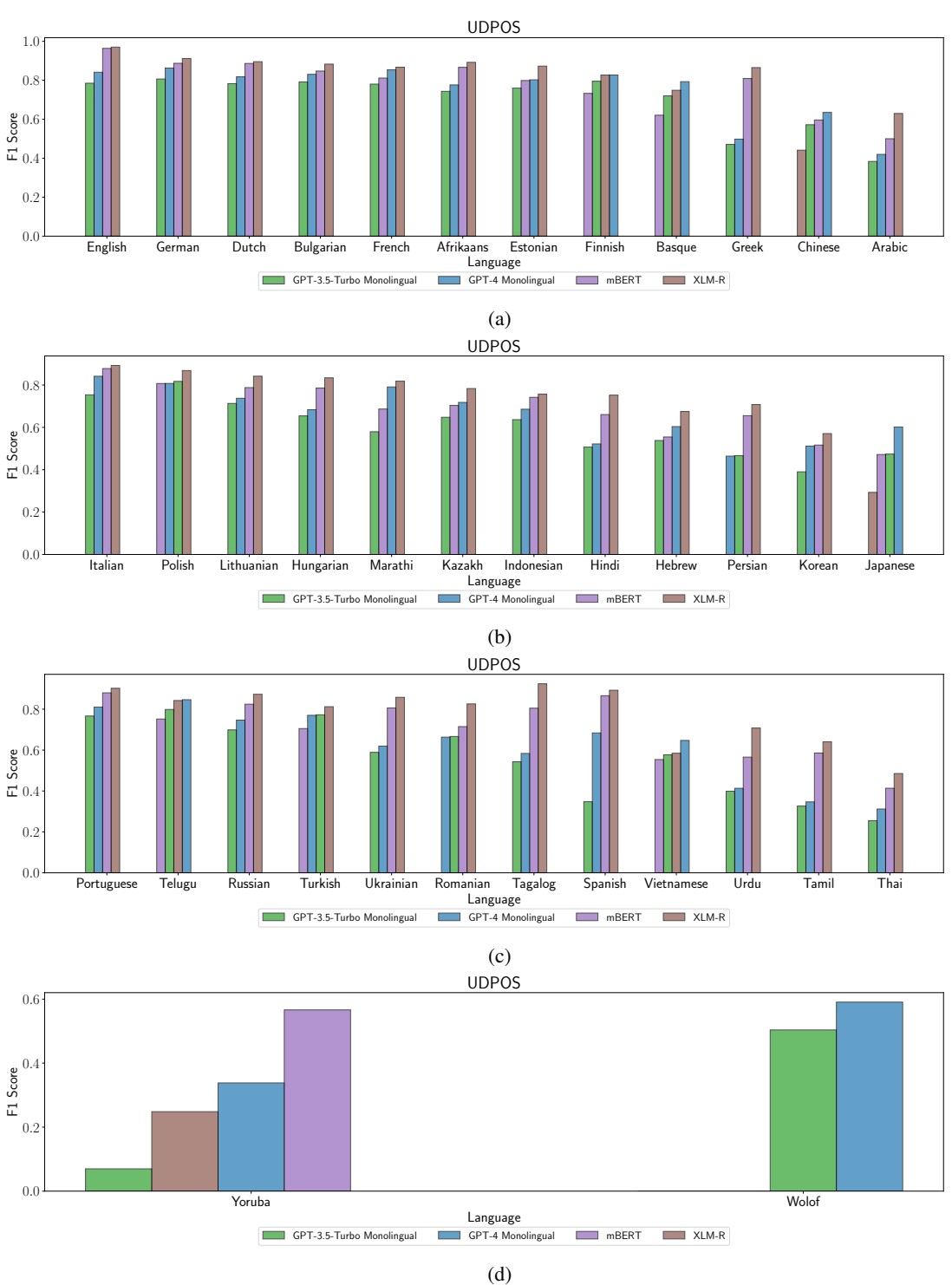

Figure 13: Comparing performance of different models on UDPOS

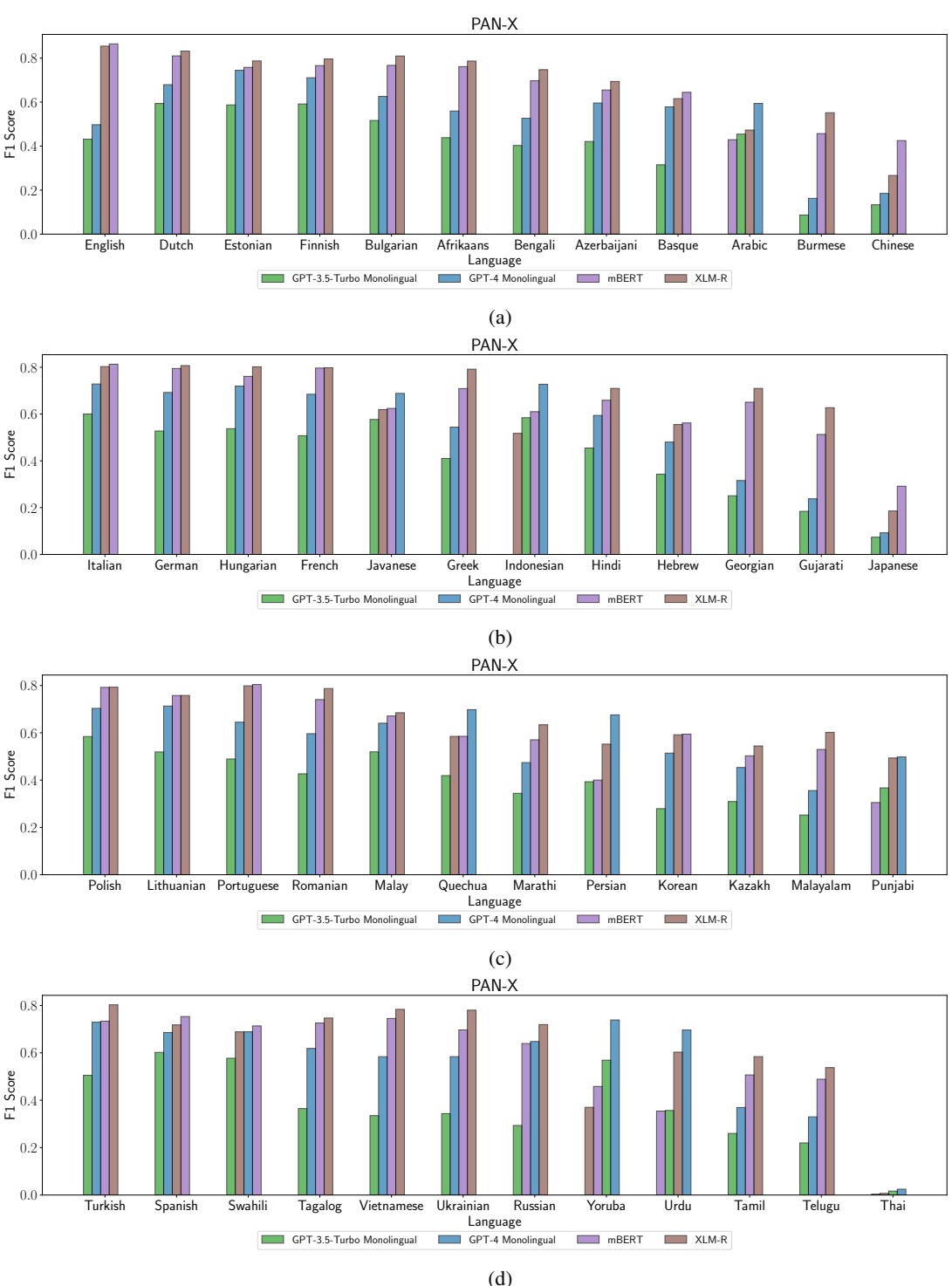

Figure 14: Comparing performance of different models on PAN-X

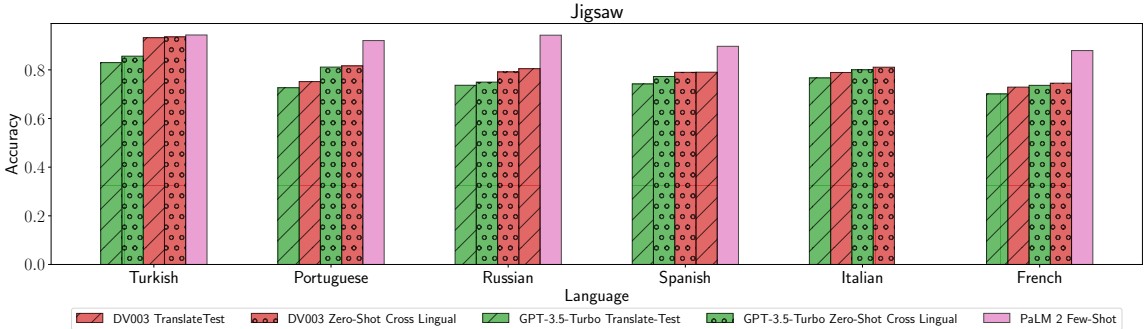

Figure 15: Comparing performance of different models on the Jigsaw dataset.

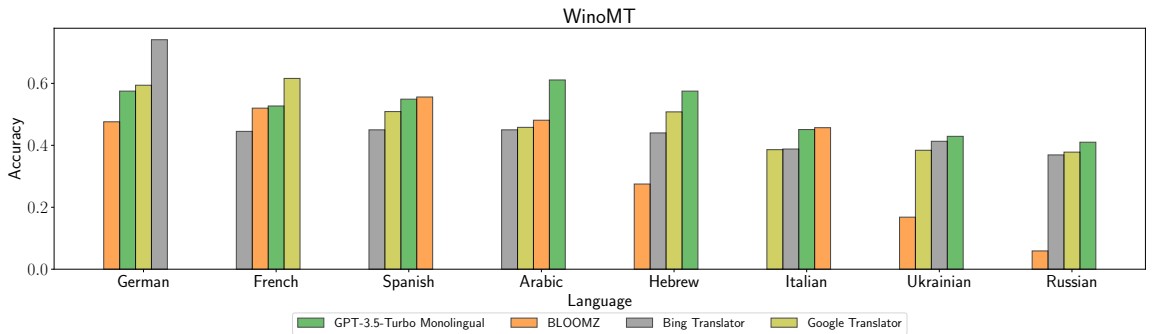

Figure 16: Comparing performance of different models on the WinoMT dataset.

| Model | en | ar | bg | de | el | es | fr | hi | ru | sw | th | tr | ur | vi | zh | **avg** |
|---|---|---|---|---|---|---|---|---|---|---|---|---|---|---|---|---|
| *Fine-tuned Baselines* | | | | | | | | | | | | | | | | |
| mBERT | 80.8 | 64.3 | 68.0 | 70.0 | 65.3 | 73.5 | 73.4 | 58.9 | 67.8 | 49.7 | 54.1 | 60.9 | 57.2 | 69.3 | 67.8 | 65.4 |
| mT5-Base | 84.7 | 73.3 | 78.6 | 77.4 | 77.1 | 80.3 | 79.1 | 70.8 | 77.1 | 69.4 | 73.2 | 72.8 | 68.3 | 74.2 | 74.1 | 75.4 |
| XLM-R Large | 88.7 | 77.2 | 83.0 | 82.5 | 80.8 | 83.7 | 82.2 | 75.6 | 79.1 | 71.2 | 77.4 | 78.0 | 71.7 | 79.3 | 78.2 | 79.2 |
| TuLRv6 - XXL | **93.3** | **89.0** | **90.6** | **90.0** | **90.2** | **91.1** | **90.7** | **86.2** | **89.2** | **85.5** | **87.5** | **88.4** | **82.7** | **89.0** | **88.4** | **88.8** |
| *Prompt-Based Baselines* | | | | | | | | | | | | | | | | |
| BLOOMZ | 67.5 | 60.7 | 46.5 | 54.0 | 47.4 | 61.2 | 61.4 | 56.8 | 53.3 | 50.4 | 43.8 | 42.7 | 50.0 | 61.0 | 56.7 | 54.2 |
| XGLM | 52.6 | 46.4 | 48.9 | 45.6 | 48.7 | 45.8 | 49.4 | 46.8 | 48.6 | 44.5 | 46.6 | 45.4 | 43.4 | 48.5 | 48.8 | 47.3 |
| *Open AI Models* | | | | | | | | | | | | | | | | |
| gpt-3.5-turbo | 76.2 | 59.0 | 63.5 | 67.3 | 65.1 | 70.3 | 67.7 | 55.5 | 62.5 | 56.3 | 54.0 | 62.6 | 49.1 | 60.9 | 62.1 | 62.1 |
| gpt-3.5-turbo (TT) | 76.2 | 62.7 | 67.3 | 69.4 | 67.2 | 69.6 | 69.0 | 59.9 | 63.7 | 55.8 | 59.6 | 63.8 | 54.0 | 63.9 | 62.6 | 64.3 |
| text-davinci-003 | 79.5 | 52.2 | 61.8 | 65.8 | 59.7 | 71.0 | 65.7 | 47.6 | 62.2 | 50.2 | 51.1 | 57.9 | 50.0 | 56.4 | 58.0 | 59.3 |
| text-davinci-003 (TT) | 79.5 | 65.1 | 70.8 | 71.7 | 69.3 | 72.2 | 71.8 | 63.3 | 67.3 | 57.3 | 62.0 | 67.6 | 55.1 | 66.9 | 65.8 | 67.1 |
| gpt-4-32k | 84.9 | 73.1 | 77.3 | 78.8 | 79.0 | 78.8 | 79.5 | 72.0 | 74.3 | 70.9 | 68.8 | 76.3 | 68.1 | 74.3 | 74.6 | 75.4 |

Table 9: Comparing performance of different models on all languages in XNLI. Metric: Accuracy.

| Model | as | bn | gu | hi | kn | ml | mr | or | pa | ta | te | **avg** |
|---|---|---|---|---|---|---|---|---|---|---|---|---|
| *Fine-tuned Baselines* | | | | | | | | | | | | |
| MuRIL | **76.0** | **75.0** | **77.0** | **77.0** | **77.0** | **79.0** | **74.0** | **76.0** | **77.0** | **77.0** | **74.0** | **76.0** |
| *Open AI Models* | | | | | | | | | | | | |
| gpt-3.5-turbo | 49.5 | 53.6 | 50.6 | 55.5 | 53.9 | 48.4 | 49.9 | 47.4 | 53.6 | 48.2 | 47.4 | 50.7 |
| gpt-3.5-turbo (TT) | 54.3 | 61.6 | 61.8 | 59.6 | 60.8 | 59.9 | 58.7 | 58.5 | 62.3 | 58.3 | 60.8 | 59.7 |
| text-davinci-003 | 48.6 | 52.6 | 51.2 | 56.9 | 49.1 | 48.2 | 49.4 | 46.4 | 50.4 | 45.5 | 47.2 | 49.6 |
| text-davinci-003 (TT) | 56.0 | 66.0 | 64.7 | 62.6 | 63.9 | 61.8 | 60.9 | 60.8 | 64.7 | 61.8 | 63.1 | 62.4 |
| gpt-4-32k | 63.5 | 72.2 | 66.9 | 71.7 | 69.0 | 64.3 | 66.2 | 61.1 | 71.1 | 63.7 | 64.8 | 66.8 |

Table 10: Comparing performance of different models on all languages in IndicXNLI. Metric: Accuracy.

| Model | en | de | es | fr | ja | ko | zh | **avg** |
|---|---|---|---|---|---|---|---|---|
| *Fine-tuned Baselines* | | | | | | | | |
| mBERT | 94.0 | 85.7 | 87.4 | 87.0 | 73.0 | 69.6 | 77.0 | 81.9 |
| mT5-Base | 95.4 | 89.4 | 89.6 | 91.2 | 79.8 | 78.5 | 81.1 | 86.4 |
| XLM-R Large | 94.7 | 89.7 | 90.1 | 90.4 | 78.7 | 79.0 | 82.3 | 86.4 |
| TuLRv6 - XXL | **97.2** | **95.1** | **94.8** | **95.6** | **89.4** | **90.4** | **90.4** | **93.2** |
| *Prompt-Based Baselines* | | | | | | | | |
| BLOOMZ | 89.8 | 84.3 | 88.9 | 87.5 | 74.4 | 85.8 | 65.2 | 82.3 |
| *Open AI Models* | | | | | | | | |
| gpt-3.5-turbo | 72.4 | 70.6 | 72.0 | 72.1 | 67.2 | 66.5 | 69.2 | 70.0 |
| gpt-3.5-turbo (TT) | 72.4 | 70.8 | 69.7 | 70.1 | 61.9 | 62.5 | 63.1 | 67.2 |
| text-davinci-003 | 72.5 | 70.6 | 72.7 | 70.7 | 60.6 | 61.8 | 60.8 | 67.1 |
| text-davinci-003 (TT) | 72.5 | 69.8 | 70.1 | 71.3 | 65.4 | 65.8 | 65.2 | 68.6 |
| gpt-4-32k | 76.2 | 74.0 | 74.1 | 72.6 | 71.5 | 69.9 | 72.6 | 73.0 |

Table 11: Comparing performance of different models on all languages in PAWS-X. Metric: Accuracy.

| Model | en | et | ht | id | it | qu | sw | ta | th | tr | **avg** |
|---|---|---|---|---|---|---|---|---|---|---|---|
| *Fine-tuned Baselines* | | | | | | | | | | | |
| mT5-Base | - | 50.3 | 49.9 | 49.2 | 49.6 | 50.5 | 50.4 | 49.2 | 50.7 | 49.5 | 49.9 |
| TuLRv6 - XXL | - | 77.4 | 78.0 | 92.6 | 96.0 | 61.0 | 69.4 | 85.4 | 87.2 | 92.8 | 74.0 |
| *Prompt-Based Baselines* | | | | | | | | | | | |
| BLOOMZ | 88.0 | 48.0 | 55.0 | 86.0 | 74.0 | 50.0 | 60.0 | 67.0 | 50.0 | 54.0 | 63.2 |
| XGLM | - | 65.9 | 58.9 | 68.9 | 69.2 | 47.1 | 62.9 | 56.3 | 62.0 | 58.5 | 61.1 |
| *Open AI Models* | | | | | | | | | | | |
| gpt-3.5-turbo | 97.8 | 90.6 | 72.0 | 90.4 | 95.2 | 54.6 | 82.0 | 59.0 | 77.6 | 91.0 | 81.0 |
| gpt-3.5-turbo (TT) | 97.8 | 88.2 | 79.4 | 90.8 | 94.4 | 50.0 | 77.6 | 87.0 | 82.2 | 87.8 | 83.5 |
| text-davinci-003 | 98.2 | 87.8 | 75.0 | 91.4 | 96.0 | 54.8 | 63.6 | 53.8 | 66.6 | 87.8 | 77.5 |
| text-davinci-003 (TT) | 98.2 | 89.6 | 82.8 | 93.0 | 94.6 | 50.0 | 82.8 | 87.0 | 84.8 | 89.8 | 85.3 |
| gpt-4-32k | **99.6** | **98.8** | **93.2** | **97.6** | **99.8** | 58.6 | **94.4** | 79.6 | **87.8** | **97.4** | **90.7** |
| gpt-4-32k (TT) | **99.6** | 94.4 | 85.8 | 96.0 | 98.2 | **85.8** | 83.4 | **91.4** | **87.8** | 92.2 | 90.6 |

Table 12: Comparing performance of different models on all languages in XCOPA. Metric: Accuracy.

| Model | en | ar | de | el | es | hi | ru | th | tr | vi | zh | **avg** |
|---|---|---|---|---|---|---|---|---|---|---|---|---|
| *Fine-tuned Baselines* | | | | | | | | | | | | |
| mBERT | 83.5 / 72.2 | 61.5 / 45.1 | 70.6 / 54.0 | 62.6 / 44.9 | 75.5 / 56.9 | 59.2 / 46.0 | 71.3 / 53.3 | 42.7 / 33.5 | 55.4 / 40.1 | 69.5 / 49.6 | 58.0 / 48.3 | 64.5 / 49.4 |
| mT5-Base | 84.6 / 71.7 | 63.8 / 44.3 | 73.8 / 54.5 | 59.6 / 35.6 | 74.8 / 56.1 | 60.3 / 43.4 | 57.8 / 34.7 | 57.6 / 45.7 | 67.9 / 48.2 | 70.7 / 50.3 | 66.1 / 54.1 | 67.0 / 49.0 |
| XLM-R Large | 86.5 / 75.7 | 68.6 / 49.0 | 80.4 / 63.4 | 79.8 / 61.7 | 82.0 / 63.9 | 76.7 / 59.7 | 80.1 / 64.3 | 74.2 / 62.8 | 75.9 / 59.3 | 79.1 / 59.0 | 59.3 / 50.0 | 76.6 / 60.8 |
| TuLRv6 - XXL | 90.1 / 80.6 | **85.4 / 69.6** | **86.1 / 70.4** | **86.3 / 70.4** | **87.6 / 71.0** | **85.9 / 70.5** | **86.8 / 73.2** | **87.0 / 81.1** | **84.3 / 71.0** | **87.6 / 71.3** | 79.2 / 73.2 | **86.0 / 72.9** |
| *Prompt-Based Baselines* | | | | | | | | | | | | |
| BLOOMZ | **92.1 / 83.8** | 82.8 / 69.7 | 76.3 / 60.4 | 49.7 / 37.6 | 86.8 / 71.4 | 83.4 / 72.9 | 65.7 / 47.2 | 20.5 / 15.5 | 51.4 / 37.2 | 86.9 / 72.7 | **82.4 / 78.6** | 70.7 / 58.8 |
| *Open AI Models* | | | | | | | | | | | | |
| gpt-3.5-turbo | 79.3 / 58.7 | 59.6 / 35.1 | 70.6 / 46.6 | 49.0 / 22.8 | 70.3 / 40.8 | 54.0 / 29.0 | 58.0 / 31.3 | 41.9 / 30.4 | 61.8 / 35.0 | 69.1 / 42.4 | 50.4 / 48.3 | 60.4 / 38.2 |
| text-davinci-003 | 77.2 / 61.8 | 36.8 / 22.5 | 55.2 / 39.7 | 31.8 / 19.7 | 61.8 / 41.3 | 19.9 / 10.0 | 29.4 / 17.6 | 11.5 / 8.7 | 44.8 / 29.2 | 41.7 / 25.4 | 35.6 / 32.8 | 40.5 / 28.1 |
| gpt-4-32k | 83.2 / 65.6 | 67.8 / 42.4 | 71.9 / 48.7 | 62.3 / 36.6 | 77.5 / 50.7 | 63.9 / 36.7 | 63.8 / 35.8 | 54.6 / 42.0 | 70.8 / 46.6 | 75.8 / 49.7 | 60.0 / 57.5 | 68.3 / 46.6 |

Table 13: Comparing performance of different models on all languages in XQuAD. Metric: F1 Score / Exact Match.

| Model | en | ar | bn | fi | id | ko | ru | sw | te | **avg** |
|---|---|---|---|---|---|---|---|---|---|---|
| *Fine-tuned Baselines* | | | | | | | | | | |
| mBERT | 75.3 / 63.6 | 62.2 / 42.8 | 49.3 / 32.7 | 59.7 / 45.3 | 64.8 / 45.8 | 58.8 / 50.0 | 60.0 / 38.8 | 57.5 / 37.9 | 49.6 / 38.4 | 59.7 / 43.9 |
| mT5-Base | 71.8 / 60.9 | 67.1 / 50.4 | 40.7 / 22.1 | 67.0 / 52.2 | 71.3 / 54.5 | 49.5 / 37.7 | 54.9 / 32.6 | 60.4 / 43.9 | 40.6 / 31.1 | 58.1 / 42.8 |
| XLM-R Large | 71.5 / 56.8 | 67.6 / 40.4 | 64.0 / 47.8 | 70.5 / 53.2 | 77.4 / 61.9 | 31.9 / 10.9 | 67.0 / 42.1 | 66.1 / 48.1 | 70.1 / 43.6 | 65.1 / 45.0 |
| TuLRv6 - XXL | **85.4 / 76.4** | **84.1 / 70.4** | 86.9 / 79.6 | **83.8 / 72.8** | **88.8 / 77.9** | **78.5 / 67.8** | **81.9 / 68.6** | **87.2 / 79.6** | 85.2 / 71.6 | **84.6 / 73.8** |
| *Prompt-Based Baselines* | | | | | | | | | | |
| BLOOMZ | 82.4 / 70.9 | 81.9 / 62.2 | **87.8 / 82.3** | 43.6 / 28.6 | 85.0 / 71.0 | 52.3 / 43.1 | 67.4 / 51.5 | 86.0 / 77.2 | **90.3 / 81.6** | 75.2 / 63.2 |
| *Open AI Models* | | | | | | | | | | |
| gpt-3.5-turbo | 54.8 / 30.7 | 50.9 / 24.2 | 60.7 / 32.7 | 66.6 / 49.0 | 67.2 / 43.4 | 59.7 / 45.3 | 45.8 / 20.0 | 64.3 / 47.7 | 70.9 / 53.1 | 60.1 / 38.4 |
| text-davinci-003 | 73.7 / 59.1 | 56.2 / 38.7 | 16.1 / 10.6 | 70.3 / 58.8 | 68.6 / 51.2 | 40.6 / 32.2 | 42.3 / 28.9 | 74.1 / 62.3 | 5.8 / 3.0 | 49.8 / 38.3 |
| gpt-4-32k | 72.9 / 51.4 | 60.8 / 32.7 | 68.0 / 42.5 | 75.4 / 57.7 | 80.8 / 61.1 | 69.7 / 58.5 | 61.4 / 30.5 | 81.8 / 68.7 | 72.5 / 54.9 | 71.5 / 50.9 |

Table 14: Comparing performance of different models on all languages in TyDiQA. Metric: F1 Score / Exact Match.

| Model | en | ar | de | es | hi | vi | zh | **avg** |
|---|---|---|---|---|---|---|---|---|
| *Fine-tuned Baselines* | | | | | | | | |
| mBERT | 80.2 / 67.0 | 52.3 / 34.6 | 59.0 / 43.8 | 67.4 / 49.2 | 50.2 / 35.3 | 61.2 / 40.7 | 59.6 / 38.6 | 61.4 / 44.2 |
| mT5-Base | 81.7 / 66.9 | 57.1 / 36.9 | 62.1 / 43.2 | 67.1 / 47.2 | 55.4 / 37.9 | 65.9 / 44.1 | 61.6 / 38.6 | 64.4 / 45.0 |
| XLM-R Large | 83.5 / 70.6 | 66.6 / 47.1 | 70.1 / 54.9 | 74.1 / 56.6 | 70.6 / 53.1 | 74.0 / 52.9 | 62.1 / 37.0 | 71.6 / 53.2 |
| TuLRv6 - XXL | **86.6 / 74.4** | **76.2 / 56.5** | **80.2 / 67.0** | **81.7 / 65.1** | **82.2 / 64.8** | **82.3 / 63.2** | **78.1 / 56.5** | **81.0 / 63.9** |
| *Open AI Models* | | | | | | | | |
| gpt-3.5-turbo | 72.8 / 53.2 | 48.5 / 23.9 | 51.0 / 29.6 | 53.8 / 29.4 | 50.7 / 28.9 | 58.9 / 35.1 | 56.7 / 29.4 | 56.1 / 32.8 |
| gpt-3.5-turbo (TT) | 72.8 / 53.2 | 37.8 / 18.4 | 44.3 / 26.2 | 54.1 / 31.8 | 37.3 / 20.0 | 41.6 / 22.5 | 36.5 / 17.2 | 46.4 / 27.0 |
| text-davinci-003 | 74.8 / 59.0 | 38.4 / 21.7 | 57.7 / 38.1 | 62.9 / 37.8 | 24.9 / 14.1 | 47.7 / 29.7 | 32.3 / 31.7 | 48.4 / 33.1 |
| text-davinci-003 (TT) | 74.8 / 59.0 | 48.2 / 25.6 | 53.5 / 33.9 | 62.9 / 40.9 | 49.2 / 28.7 | 51.0 / 30.4 | 45.2 / 24.1 | 55.0 / 34.7 |
| gpt-4-32k | 80.3 / 62.8 | 59.1 / 33.5 | 64.7 / 44.4 | 70.0 / 45.9 | 57.3 / 35.6 | 72.2 / 49.0 | 67.1 / 38.4 | 67.2 / 44.2 |

Table 15: Comparing performance of different models on all languages in MLQA. Metric: F1 Score / Exact Match.

| Model | as | bn | gu | hi | kn | ml | mr | or | pa | ta | te | **avg** |
|---|---|---|---|---|---|---|---|---|---|---|---|---|
| *Fine-tuned Baselines* | | | | | | | | | | | | |
| BLOOMZ | 40.6 / 31.7 | 42.9 / 36.6 | 37.2 / 29.9 | 44.0 / 45.1 | 37.8 / 26.6 | 30.5 / 28.4 | 39.2 / 33.0 | 25.4 / 22.0 | 26.4 / 33.5 | 39.7 / 35.9 | 38.9 / 34.7 | 36.6 / 32.5 |
| *Open AI Models* | | | | | | | | | | | | |
| gpt-3.5-turbo | 35.3 / 21.4 | 49.5 / 30.2 | 40.5 / 25.5 | 55.9 / 39.3 | 35.3 / 20.4 | 30.0 / 19.2 | 50.0 / 32.0 | 22.1 / 12.7 | 35.8 / 15.1 | 32.7 / 21.6 | 32.9 / 19.7 | 38.2 / 23.4 |
| text-davinci-003 | 6.7 / 3.2 | 10.3 / 5.8 | 5.4 / 3.5 | 16.8 / 11.8 | 7.1 / 3.9 | 3.6 / 2.3 | 14.6 / 8.5 | 6.9 / 3.4 | 10.7 / 4.1 | 4.2 / 2.5 | 6.8 / 3.6 | 8.4 / 4.8 |
| gpt-4-32k | **58.8 / 40.4** | **67.1 / 47.4** | **59.4 / 42.4** | **75.2 / 62.2** | **47.1 / 31.6** | **48.3 / 33.7** | **60.7 / 43.1** | **29.9 / 16.7** | **56.1 / 34.1** | **54.0 / 39.7** | **47.9 / 27.8** | **55.0 / 38.1** |

Table 16: Comparing performance of different models on all languages in IndicQA. Metric: F1 Score / Exact Match.

| Model | en | af | ar | bg | de | el | es | et | eu | fa | fi | fr | he | hi | hu | id | it | ja | kk |
|---|---|---|---|---|---|---|---|---|---|---|---|---|---|---|---|---|---|---|---|
| *Fine-tuned Baselines* | | | | | | | | | | | | | | | | | | | |
| mBERT | 96.4 | 86.7 | 50.0 | 84.7 | 88.7 | 80.9 | 86.6 | 79.9 | 62.1 | 65.5 | 73.3 | 81.2 | 55.5 | 66.0 | 78.6 | 74.2 | 87.8 | 47.2 | 70.4 |
| XLM-R Large | **97.0** | **89.2** | **63.0** | **88.3** | **91.2** | **86.5** | **89.2** | **87.3** | 74.9 | **70.8** | 82.7 | **86.7** | 67.5 | 75.2 | 83.4 | 75.7 | **89.2** | 29.3 | **78.3** |
| *Open AI Models* | | | | | | | | | | | | | | | | | | | |
| gpt-3.5-turbo | 78.5 | 74.3 | 38.3 | 79.1 | 80.7 | 47.1 | 34.8 | 76.0 | 72.0 | 46.7 | 79.5 | 78.0 | 53.8 | 50.7 | 65.4 | 63.6 | 75.4 | 47.4 | 64.8 |
| gpt-4-32k | 84.1 | 77.6 | 42.0 | 83.1 | 86.3 | 49.8 | 68.4 | 80.2 | **79.3** | 46.4 | **82.7** | 85.4 | 60.4 | 52.2 | 68.3 | 68.6 | 84.1 | **60.2** | 71.8 |

| | ko | lt | mr | nl | pl | pt | ro | ru | ta | te | th | tl | tr | uk | ur | vi | wo | yo | zh | **avg** |
|---|---|---|---|---|---|---|---|---|---|---|---|---|---|---|---|---|---|---|---|---|
| *Fine-tuned Baselines* | | | | | | | | | | | | | | | | | | | | |
| mBERT | 51.7 | 78.8 | 68.7 | 88.6 | 80.7 | 88.0 | 71.5 | 82.4 | 58.5 | 75.2 | 41.3 | 80.5 | 70.5 | 80.6 | 56.6 | 55.4 | 0.0 | **56.6** | **59.6** | 71.9 |
| XLM-R Large | 57.1 | **84.2** | **81.8** | 89.5 | 86.8 | 90.2 | 82.6 | 87.3 | 64.0 | 84.2 | 48.5 | **92.4** | 81.2 | 85.8 | 70.8 | 58.5 | 0.0 | 24.8 | 44.1 | **76.2** |
| *Open AI Models* | | | | | | | | | | | | | | | | | | | | |
| gpt-3.5-turbo | 39.0 | 71.3 | 57.9 | 78.3 | 81.7 | 76.7 | 66.7 | 69.9 | 32.6 | 79.8 | 25.5 | 54.3 | 77.2 | 58.9 | 39.9 | 57.7 | 50.4 | 7.0 | 57.2 | 60.2 |
| gpt-4-32k | 51.2 | 73.7 | 79.1 | 81.8† | 80.7 | 81.0 | 66.3† | 74.7 | 34.7 | **84.6** | 31.2† | 58.4† | 77.0 | 61.9 | 41.3 | **64.7** | 59.1 | 33.8† | 63.5 | 66.6 |

Table 17: Comparing performance of different models on all languages in POS. Metric: F1 Score. (All numbers are Monolingual results except the ones marked with † symbol which indicate Zero-Shot Cross-Lingual results (due to the absence of training data in those languages)

| Model | en | af | ar | az | bg | bn | de | el | es | et | eu | fa | fi | fr | gu | he | hi | hu | id | it | ja | jv | ka | kk |
|---|---|---|---|---|---|---|---|---|---|---|---|---|---|---|---|---|---|---|---|---|---|---|---|---|
| *Fine-tuned Baselines* | | | | | | | | | | | | | | | | | | | | | | | | |
| mBERT | **86.4** | 76.1 | 42.9 | 65.5 | 76.7 | 69.7 | 79.5 | 70.9 | **75.3** | 75.8 | **64.4** | 40.0 | 76.6 | 79.6 | 51.3 | **56.2** | 65.9 | 76.1 | 61.0 | **81.3** | 29.2 | 62.4 | 65.1 | 50.3 |
| XLM-R Large | 85.4 | **78.6** | 47.3 | **69.4** | 80.9 | 74.7 | 80.7 | 79.2 | 71.8 | **78.7** | 61.6 | 55.2 | **79.6** | 79.8 | 62.7 | 55.5 | **70.9** | 80.2 | 51.8 | 80.3 | 18.5 | 61.9 | **70.9** | 54.4 |
| *Open AI Models* | | | | | | | | | | | | | | | | | | | | | | | | |
| gpt-3.5-turbo | 43.2 | 43.8 | 45.4 | 42.1 | 51.6 | 40.3 | 52.7 | 41.0 | 60.2 | 58.7 | 31.5 | 39.3 | 59.1 | 50.7 | 18.4 | 34.3 | 45.5 | 53.7 | 58.4 | 60.0 | 7.4 | 57.7 | 25.1 | 30.9 |
| gpt-4-32k | 49.7 | 55.9 | **59.4** | 59.6 | 62.6 | 52.7 | 69.2 | 54.4 | 68.6 | 74.4 | 57.8 | **67.6** | 71.1 | 68.5 | 23.8 | 48.0 | 59.4 | 71.9 | **72.7** | 72.8 | 9.2 | **68.8** | 31.6 | 45.3 |

| | ko | lt | ml | mr | ms | my | nl | pa | pl | pt | qu | ro | ru | sw | ta | te | th | tl | tr | uk | ur | vi | yo | zh | **avg** |
|---|---|---|---|---|---|---|---|---|---|---|---|---|---|---|---|---|---|---|---|---|---|---|---|---|---|
| *Fine-tuned Baselines* | | | | | | | | | | | | | | | | | | | | | | | | | |
| mBERT | 59.5 | **75.8** | 53.0 | 57.0 | 67.1 | 45.7 | 81.0 | 30.5 | 79.2 | **80.4** | 58.5 | 74.0 | 63.9 | **71.4** | 50.7 | 48.9 | 0.4 | 72.6 | 73.4 | 69.7 | 35.4 | 74.5 | 45.8 | **42.5** | 62.3 |
| XLM-R Large | 59.2 | **75.8** | 60.2 | 63.4 | 68.5 | 55.2 | 83.2 | 49.4 | **79.3** | 79.9 | 58.5 | **78.7** | 71.9 | 68.9 | 58.4 | 53.8 | 0.7 | 74.7 | 80.3 | 78.0 | 60.3 | **78.3** | 37.0 | 26.6 | **65.2** |
| *Open AI Models* | | | | | | | | | | | | | | | | | | | | | | | | | |
| gpt-3.5-turbo | 27.9 | 51.9 | 25.2 | 34.4 | 52.0 | 8.7 | 59.4 | 36.7 | 58.4 | 48.9 | 41.9 | 42.7 | 29.4 | 57.7 | 26.0 | 22.0 | 1.7 | 36.5 | 50.5 | 34.4 | 35.7 | 33.5 | 56.9 | 13.3 | 40.3 |
| gpt-4-32k | 51.4 | 71.3 | 35.6 | 47.4 | 64.1 | 16.3 | 67.9 | 49.8 | 70.3 | 64.5 | **69.8** | 59.6 | 64.8 | 68.9 | 36.9 | 33.0 | **2.5** | 61.9 | 72.9 | 58.4 | **69.6** | 58.4 | **73.9** | 18.5 | 55.5 |

Table 18: Comparing performance of different models on all languages in PAN-X. Metric: F1 Score.

| Model | ar | en | es | eu | hi | id | my | ru | sw | te | zh | **avg** |
|---|---|---|---|---|---|---|---|---|---|---|---|---|
| *Prompt-Based Baselines* | | | | | | | | | | | | |
| BLOOMZ | 79.7 | 95.7 | 87.3 | 70.5 | 79.9 | 85.6 | 49.9 | 67.3 | 65.3 | 67.4 | 90.0 | 76.2 |
| XGLM | 59.8 | 75.9 | 69.2 | 63.8 | 62.5 | 70.8 | 61.2 | 72.4 | 65.2 | 63.4 | 67.7 | 66.5 |
| *Open AI Models* | | | | | | | | | | | | |
| gpt-3.5-turbo | 92.5 | 96.8 | 95.8 | 78.4 | 91.1 | 95.0 | 57.2 | 96.6 | 92.3 | 73.1 | 95.6 | 87.7 |
| gpt-3.5-turbo (TT) | 94.3 | 96.8 | 96.1 | 92.5 | 94.7 | 95.2 | 88.6 | 96.2 | 88.7 | 93.6 | 95.6 | 93.9 |
| text-davinci-003 | 87.4 | 98.3 | 97.6 | 78.1 | 77.8 | 96.4 | 47.4 | 94.2 | 78.1 | 57.6 | 95.0 | 82.5 |
| text-davinci-003 (TT) | 95.0 | 98.3 | 96.2 | 94.1 | 95.1 | 95.9 | 90.1 | 96.9 | 90.7 | 94.3 | 96.2 | 94.8 |
| gpt-4-32k | **99.1** | **99.6** | **99.5** | **97.6** | **98.8** | **99.0** | 77.6 | 99.1 | **98.4** | 93.4 | **99.2** | 96.5 |
| gpt-4-32k (TT) | 97.7 | **99.6** | 98.7 | 96.8 | 97.9 | 98.1 | **93.2** | **99.2** | 93.6 | **96.4** | 98.3 | **97.0** |

Table 19: Comparing performance of different models on all languages in XStoryCloze. Metric: Accuracy.

| | Google | | | Microsoft | | | Amazon | | | Systran | | | GPT Turbo 3.5 | | | Bloomz | | |
|----|------|------------|------------|------|------------|------------|------|------------|------------|------|------------|------------|------|------------|------------|------|------------|------------|
| | Acc | $\Delta_G$ | $\Delta_S$ | Acc | $\Delta_G$ | $\Delta_S$ | Acc | $\Delta_G$ | $\Delta_S$ | Acc | $\Delta_G$ | $\Delta_S$ | Acc | $\Delta_G$ | $\Delta_S$ | Acc | $\Delta_G$ | $\Delta_S$ |
| es | 50.9 | 23.2 | 20.9 | 45 | 36.5 | 22.9 | **57.2** | 15.3 | 21.7 | 42.5 | 46.2 | 15.6 | 54.9 | 22.7 | 26.2 | 55.6 | 17.2 | 32.5 |
| fr | **61.6** | 6.1 | 22.3 | 44.5 | 34.2 | 15.8 | 54.2 | 16.4 | 15 | 43.4 | 41.8 | -0.1 | 52.7 | 21.4 | 26.1 | 52 | 17.8 | 24.6 |
| it | 38.6 | 32.9 | 18.6 | 38.8 | 41.8 | 10.5 | 40.2 | 26.8 | 14.7 | 38.1 | 47.3 | 6.3 | 45.1 | 21.9 | 26.7 | **45.7** | 9 | 18.5 |
| ru | 37.8 | 36.7 | 11.4 | 36.9 | 42 | 8.4 | 39.8 | 34.8 | 9.4 | 37.3 | 44.1 | 9.2 | **41** | 31.6 | 10.2 | 5.9 | INV | 0 |
| uk | 38.4 | 43.5 | 10.7 | 41.3 | 46.8 | 11.9 | - | - | - | 28.9 | 22.4 | 12.9 | **42.9** | 34.2 | 12.1 | 16.8 | 22.7 | 2.2 |
| he | 50.8 | 11.7 | 35.5 | 44 | 22 | 29.8 | 48 | 13.6 | 45.9 | 43.1 | 26.9 | 23.1 | **57.5** | 7.6 | 40.8 | 27.5 | 31.4 | 5 |
| ar | 45.8 | 42.5 | 16.2 | 45 | 47.1 | 14.2 | 48.3 | 37.8 | 18.8 | 45.6 | 49.4 | -4.1 | **61.1** | 13.9 | 27.9 | 48.1 | 23 | 25.6 |
| de | 59.4 | 12.5 | 12.6 | **74.1** | 0 | 8.8 | 62.4 | 12 | 16.7 | 48.5 | 34.5 | 10 | 57.5 | 19.5 | 14.2 | 47.6 | 56.2 | 6.6 |

Table 20: Performance of commercial MT systems and LLMs on the WinoMT corpus on 8 target languages. Results are categorized by language family. Acc indicates overall gender accuracy (% of instances the translation had the correct gender), $\Delta_G$ denotes the difference in performance (F1 score) between masculine and feminine scores, and $\Delta_S$ is the difference in performance (F1 score) between pro-stereotypical and anti-stereotypical gender role assignments (higher numbers in the two latter metrics indicate stronger biases). Numbers in bold indicate best accuracy for the language across all systems. Notes: [1. For Google, Microsoft, Amazon, and Systran we use the translations provided by (Stanovsky et al., 2019). Some values differ from the original paper due to updated Spcay modules. 2. For Ru in Bloomz, Precision in male predictions is 0 leading to Invalid (INV) in $\Delta_G$]

| Model | es | fr | it | pt | ru | tr | **avg** |
|-------|------|------|------|------|------|------|------|
| *LLM Baselines* | | | | | | | |
| PALM (0-Shot) | 79.83 | 78.99 | - | 77.58 | 80.35 | 84.1 | 80.17 |
| PALM (10-Shot Monolingual) | **91.23** | 86.16 | - | 90.99 | 92.47 | 84.5 | 89.07 |
| PALM-2 (0-Shot) | 88.6 | 84.11 | - | 87.68 | 90.5 | 93.42 | 88.86 |
| PALM-2 (10-Shot Monolingual) | 89.68 | **87.94** | - | **92.05** | **94.25** | **94.34** | **91.65** |
| *OpenAI Models* | | | | | | | |
| gpt-3.5-turbo (Crosslingual) | 77.27 | 73.64 | 80.05 | 81.16 | 74.99 | 85.65 | 78.79 |
| gpt-3.5-turbo (TT) | 74.20 | 70.09 | 76.67 | 72.66 | 73.68 | 82.99 | 75.05 |
| text-davinci-003 (Crosslingual) | 79 | 74.55 | **81.11** | 81.63 | 79.13 | 93.55 | 81.50 |
| text-davinci-003 (TT) | 79.06 | 72.93 | 78.93 | 75.18 | 80.48 | 93.22 | 79.97 |

Table 21: Comparing performance of different models on all languages in Jigsaw. Metric: Accuracy.