# OpenReview forum: "MEGA: Multilingual Evaluation of Generative AI"
_EMNLP/2023/Conference — EMNLP 2023 Main_

### Official Review · Reviewer_ppmZ · 2023-08-04

**Soundness:** 4

**Excitement:**

4: Strong: This paper deepens the understanding of some phenomenon or lowers the barriers to an existing research direction.

**Paper Topic And Main Contributions:**

The paper argues that Generative LLMs should be evaluated multilingually and provides a comprehensive benchmarking.

The precise research questions are: (1), how well do LLMs fare on multilingual benchmarks compared to fine-tuned  SOTA models? (2), what languages do these models perform well in, and can we explain the trends in performance for these models across languages? (3), what prompting strategies should be used for  using LLMs for non-English languages?


I think this paper provides a convincing contribution (in terms of models tested and language diversity of their 70 languages)  to a core problem that could have real impact.


**Questions For The Authors:**

A: There is a fourth research question that you did not investigate that is : how much English is used internally really? You acknowledge it (lines 104-108) but do not handle it. Why is that, and why don’t you think that this could undermine your entire enterprise?
Can you propose tasks that would discover translations from English?

B: Figure 2 is not clear, what is the y-axis?

C: I find your results very interesting, especially those on memorisation, could you extend the discussion? For example, what does it mean given that you demonstrate that data is contaminated? How serious is the problem and how does it relate to the multilingual aspect? What should be done to avoid that?


**Reasons To Accept:**

The topic: The problem of honestly evaluating LLMs and separating the real advances from the hype is a central one. Multilingual evaluation should be a must  and foremost evaluation standard.

The thoroughness of the analyses.

The very interesting results, especially about data contamination.



**Reasons To Reject:**

There is a fourth question that is : how much English is used internally really? so tasks must be used that uncover translations from English. The authors themselves acknowledge this problem (lines 104-108), which to me seems pretty undermining of their whole enterprise, but do not address it in practice.

Figure 1 is far far too small, even with 200% enlargement it is still too small to read, which means the paper is more than 8 pages long. So this should be corrected.




**Reproducibility:**

4: Could mostly reproduce the results, but there may be some variation because of sample variance or minor variations in their interpretation of the protocol or method.

**Reviewer Confidence:**

4: Quite sure. I tried to check the important points carefully. It's unlikely, though conceivable, that I missed something that should affect my ratings.

**Typos Grammar Style And Presentation Improvements:**

Line 335 upto -> up to

Line 401 retrive -> retrieve

---

> ### Author Rebuttal · Authors · 2023-08-27
>
> We would like to thank the reviewer for their insightful comments and helpful feedback. Below we try to answer their questions and concerns.
>
> **How much English is used internally really?**: We are not sure if we understood the reviewer’s question here, but let us try to clarify some potential mis-understandings about the lines 104-108. In these lines, we explain that for low-resource languages, translate-test strategy for prompting i.e. first translating an example in non-English language to English (using some MT tool like google-translate or bing) and then feeding the translated example to the LLM for prediction gives the best performance. These observations are discussed in detail in Section 3.1 of the manuscript. **We didn’t mean to say that LLMs do a poor job at translation, but instead that for most cases, the best performance on non-English languages for different tasks is obtained by first translating them to English.** We agree our wording in the Lines 104-108 can be a source of confusion for the readers and we will rephrase them to convey our point properly. To the best of our understanding of the reviewer’s question, these lines do not undermine our premise and in-fact convey one of the core findings from our work. We hope this answers the reviewer's question and if we have missed something, we will be happy to discuss this further during the discussion period.
>
> **Size of Figure 1**: We agree that in the current manuscript the size of Figure 1 is small. We will ensure that with an additional page for the final version, the figure is readable at the default scale.
>
> **Y-axis in Figure 2**: We apologize for the lack of clarity here. We label the y-axis as “Score” which denotes the values for the evaluation metrics specific to each task. For eg. for XNLI the “Score” is classification accuracy and for TyDiQA-GoldP it is the F1-Score. These metrics are listed in the second row of Table 1. But we agree that since Figure 2 appears before Table 1, we should have listed them earlier and we will correct this in the final version positively. Thank you for bringing this to our attention.
>
> **Findings on Memorization**: We thank the reviewer for asking this question. Given the lack of transparency about the training details of the LLMs that we evaluate, finding concrete evidence of memorization or data contamination becomes very difficult. What we have in our work are some indicators like the model’s understanding of the task structure and accessibility and release dates of the test datasets, which convey the possibility (qualitatively) of the test data appearing in the training datasets of these LLMs. We believe that the contamination of test datasets is a serious problem for works centered around LLM evaluation (including ours), as they might lead to an overestimation of the capabilities of these models. However, we would like to highlight that despite the possibility of contamination, LLMs still vastly underperform on (especially low-resource) non-English languages . The findings on contamination would have been more troubling if our results indicated superior performance on these languages, which is not the case. In the context of our work, these observations (about data contamination) indicate that the disparity in performance between English and non-English languages (Lines 97-102), might be even greater than what we observe in our work. About the practices that can be adopted to avoid the possibility of contamination, we believe that when we construct new datasets, we can avoid the test datasets to be directly available on the web and instead could be available via some manual download policy or other licenses. We once again thank the reviewer for their questions regarding this topic and we will include this discussion in the final version of our paper.

---

### Official Review · Reviewer_svC2 · 2023-08-10

**Soundness:** 4

**Excitement:**

3: Ambivalent: It has merits (e.g., it reports state-of-the-art results, the idea is nice), but there are key weaknesses (e.g., it describes incremental work), and it can significantly benefit from another round of revision. However, I won't object to accepting it if my co-reviewers champion it.

**Paper Topic And Main Contributions:**

The paper evaluates the performance of large language models (LLMs) on various multilingual tasks and languages. The paper uses 16 tasks and 70 languages to compare LLMs with fine-tuned models. The paper also explores different ways of using prompts to elicit responses from LLMs, i.e., monolingual prompting, zero-shot cross-lingual prompting, and translate-test prompting. Through experiments, the authors find that fine-tuned models outperform large language models, especially, in low-resource settings, though techniques like translate-test prompting help bridge the gap.

**Questions For The Authors:**

Q1: How the language-specific LMs are compared with other models?

**Reasons To Accept:**

- The experiments are conducted on a wide range of tasks and languages to lead to the conclusion that the LLMs still underperform SOTA fine-tuned models.
- The analysis of different prompting strategies and their effects in the multilingual setting can offer solid baselines for other works.
- Authors carefully tune prompts.

**Reasons To Reject:**

- Authors only compare LLMs with fine-tuned multilingual pre-trained LM, it leaves open questions about other language-specific pre-trained models and how they might perform. The additional comparative analysis could be beneficial.

**Reproducibility:**

4: Could mostly reproduce the results, but there may be some variation because of sample variance or minor variations in their interpretation of the protocol or method.

**Reviewer Confidence:**

3: Pretty sure, but there's a chance I missed something. Although I have a good feel for this area in general, I did not carefully check the paper's details, e.g., the math, experimental design, or novelty.

---

> ### Author Rebuttal · Authors · 2023-08-27
>
> We would like to thank the reviewer for their insightful comments and helpful feedback. Below we try to answer their questions and concerns.
>
> **Comparison with Language-Specific Pre-trained models**: The reason we focus on multilingual pre-trained LMs only and not language-specific LMs is due to the zero-shot cross lingual properties of the multilingual models i.e. fine-tuning only with English data and then evaluating zero-shot on other languages. This is important because for most of the datasets that we study in our work, there are no language-specific training datasets that can be used for fine-tuning. Hence, in most cases these baselines are fine-tuned on English data only (Lines 204-207) and it becomes infeasible to use language specific pretrained LMs. Further, there is also evidence in prior work showing that large enough pre-trained multilingual models can outperform monolingual models pre-trained on the languages to be evaluated for [1].
>
> [1] Are the Multilingual Models Better? Improving Czech Sentiment with Transformers (Přibáň & Steinberger, RANLP 2021)

---

### Official Review · Reviewer_eRUE · 2023-08-12

**Soundness:** 3

**Excitement:**

3: Ambivalent: It has merits (e.g., it reports state-of-the-art results, the idea is nice), but there are key weaknesses (e.g., it describes incremental work), and it can significantly benefit from another round of revision. However, I won't object to accepting it if my co-reviewers champion it.

**Paper Topic And Main Contributions:**

The paper presents benchmarking results for multilingual generative models and investigates three questions:
1) how well do LLMs fare on multilingual benchmarks compared to fine-tuned SOTA models?
2) What languages do these models perform well in, and can we explain the trends in performance for these models across languages?
3) What prompting strategies should be used for using LLMs for non-English languages?

**Reasons To Accept:**

> For the languages and models provided, the paper provides good discussion and insights into the performance of these models
> Experimental prompts strategy is explained well, despite the added explanatory complexity/redundancies, and they also investigate the models for test data contamination as part of their model analysis

**Reasons To Reject:**

> The experimental section of the paper writing can be improved as it has bad or complicated writing at some parts "like ours", "overally", incomplete or jump sentences "we have the ability to follow instructions".
> The experimental section has missing information, and may not lead to easily reproducible results: How many runs is the average score? What parameters were used for fine-tuning the baseline models on the benchmark data? Does the mathematical expression in the paper add in more to the explainability? Perhaps the figure 1c is enough? Please consider these questions
> The number of languages covered by the paper models and the number of languages evaluated/described in the experimental section are two different things. The experimental section is lacking in language coverage to say this is a multilingual model benchmark paper. With more language variety considered, this can be addressed.
> The connection between the investigation questions and the paper conclusion is not very clear

For what the paper wants to represent, I believe more work needs to be done and the paper flow could be improved, by removing unnecessary complex explanation of simple concepts for instance.

**Reproducibility:**

3: Could reproduce the results with some difficulty. The settings of parameters are underspecified or subjectively determined; the training/evaluation data are not widely available.

**Reviewer Confidence:**

3: Pretty sure, but there's a chance I missed something. Although I have a good feel for this area in general, I did not carefully check the paper's details, e.g., the math, experimental design, or novelty.

**Typos Grammar Style And Presentation Improvements:**

(1, 30) Engish
(3, 214) We have the ability to follow instructions ???
(7, 472) Overally

---

> ### Author Rebuttal · Authors · 2023-08-27
>
> We would like to thank the reviewer for their insightful comments. Below we try to answer their questions and concerns.
>
> **How many runs is the average score?**: We assume that here the reviewer is talking about average scores in Tables 1 and 2, and Figure 2. Here, we have averaged the test data performance across languages available in a given dataset. We consider only a single run for evaluation, as the scale of our experiments i.e. the number of datasets and languages as well as the size of the models that we deal with, makes it computationally prohibitive to perform multiple runs for each evaluation. To ensure reproducibility we use a temperature value of 0 and we share the prompts that we use for each task and model in Section A.4 of the Appendix. We agree that some of these details are missing in the present manuscript and we will provide the same in the final version.
>
> **What parameters were used for fine-tuning the baseline models on the benchmark data?**: In all the fine-tuning baselines, we fine-tune all the parameters of the models, following other benchmarking studies like XTREME [1] that cover these models. Regarding hyperparameters used for fine-tuning, we use the hyperparameters reported in previous work [1,2]. More commonly, we used a learning rate of 3e-5, an effective batch size of 32, and 3 epochs, with some modifications depending on different models and datasets. We will list down all the hyperparameters used for fine-tuning in the final version.
>
> **Does the mathematical expression in the paper add in more to the explainability? Perhaps the figure 1c is enough?**: We firmly believe that the mathematical expression is important in understanding the inference strategy that we use for evaluating the models through prompting, especially for readers who are not well versed with the literature on prompting methods. We are able to explain different components of the prompt through this notation, which also help motivate different multilingual prompting strategies (Section 2.3.1) that we use in our work. However, we agree that in the current form this particular section is a bit long and we will make the writing more precise in the final version to make room for other experimental details.
>
> **Language Coverage**: We are not sure if we follow the phrase *“number of languages covered by the paper models”*. We provide the number of languages covered in each dataset that we evaluate in Figure 1a, where the number in parenthesis next to the name of the dataset contains the number of languages in that dataset. Further, in Figure 1b, we provide the language family distribution of the languages covered in different datasets included in MEGA. We discuss this in Lines 162-167 as well. **These are the languages that we evaluate in our work**. Due to the paucity of space we couldn’t provide the names of each language included in all of the 16 datasets in the main paper, but this information is available in Figures 7-13 or Tables 8-20 of the Appendix.
>
> **Connection between investigation questions and conclusion**: We believe that all of the three research questions that we motivate in Section 1 (Lines 89-96) are explored and answered in the paper in sufficient detail.
> - Q1. *How well do LLMs fare on multilingual benchmarks compared to fine-tuned SOTA models?*: We cover this in Section 3.2, where we answer this question by showing that LLMs lag behind fine-tuning baselines for most of the datasets and languages (373-380)
> - Q2. *What languages do these models perform well in, and can we explain the trends in performance for these models across languages?*: We discuss this in Section 3.3 of the paper, where we show that languages with little pre-training data or poor tokenizer qualities are the ones on which LLMs struggle the most to perform well. Further, in Section 3.1, we compare which languages perform well under different prompting strategies, where we show low-resource non-latin script languages benefit the most from translate-test prompting (Lines 334-338) and extremely low-resource languages suffer with zero-shot cross lingual prompting (Lines 359-366).
> - Q3. *What prompting strategies should be used for using LLMs for non-English languages?*: We cover this in Section 3.1, where we show that for low-resource and non-latin script languages translate-test works the best.
>
>  We agree that in the current manuscript, these questions are not explicitly answered in the Conclusion Section and with an additional page for the final version, we will restate these results in the Conclusion.
>
> [1] XTREME: A Massively Multilingual Multi-task Benchmark for Evaluating Cross-lingual Generalization. Hu et al. 2020.
> [2] mT5: A Massively Multilingual Pre-trained Text-to-Text Transformer. Xue et al. 2021.

---

### Official Review · Reviewer_1g3u · 2023-08-28

**Soundness:** 3

**Excitement:**

4: Strong: This paper deepens the understanding of some phenomenon or lowers the barriers to an existing research direction.

**Paper Topic And Main Contributions:**

This paper introduces MEGA (Multilingual Evaluation of Generative AI) which aims at evaluating the performances of generative LLMs in a wide variety of tasks (NLI, QA, seq. labelling, summarisation, etc.) in a wide variety of languages (70 topologically diverse languages). A wide range of models is tested, ranging from closed sourced/data models (Chat-GPT, GPT3.5, GPT4) to open models (BLOOM-like model) in a prompt-based setting, to fine-tuned-based models (mBERT, mT5, etc.).

Models are evaluated using monolingual prompting (same examples as the test language), 0-shot X-lingual prompting (example in a pivot language different from the test language), and translate-text (where test sentences are translated to English).

Overall, this very dense paper presents some nice contributions. Authors show that translating sentences from a low-resourced (LR) language into English works best (rather than leaving it in the original language). Open-AI models generally fall behind fine-tuned based models with the exception of GPT-4 which is on par with the others. Authors show performance in negatively correlated to tokenizer fertility (the latter being linked to the writing system) and to the amount of non-English data in the training data fed to the LLMs.

**Reasons To Accept:**

To the best of my knowledge, this is the first paper that aims at systematically benchmark the performance of LLMs on such a large variety of tasks and typologically diverse pool of languages.
Authors also conclude by investigating the challenges of testing LLMs and explore test data contamination, showing that there is a strong suspicion of contamination of test data for OpenAI model (indicating that their relatively good performances might only be due to having been trained on the data).

**Reasons To Reject:**

Authors could strengthen their conclusions by using additional appropriate statistical analyses to more systematically analyse the causes that explain discrepancies in the result.  Another factor of analysis (which is currently left out, but which also falls out of the scope of the paper) would be to study the the influence of the writing system.

**Reproducibility:**

3: Could reproduce the results with some difficulty. The settings of parameters are underspecified or subjectively determined; the training/evaluation data are not widely available.

**Reviewer Confidence:**

3: Pretty sure, but there's a chance I missed something. Although I have a good feel for this area in general, I did not carefully check the paper's details, e.g., the math, experimental design, or novelty.

**Typos Grammar Style And Presentation Improvements:**

It would be better to keep the bar-order in histograms in the same order, as they would be easier to compare.

---

> ### Author Rebuttal · Authors · 2023-08-28
>
> We would like to thank the reviewer for carefully going through our work and providing their valuable feedback.
>
> Regarding the statistical analysis of the trends that we observe, we would like to kindly clarify that we do not merely just eye-ball the results, but do provide a concrete analysis, especially in Section 3.3 of the paper. We measure the Pearson correlation coefficients between the observed performance (for different tasks and languages) and factors such as tokenizer fertility and pre-training size of a language. While in the main paper, we only show the results for tokenizer fertility for the tasks where the correlation is statistically significant, we do report the full results with p-values in Table 21 of the Appendix. We use phrases like *“we can partially explain this discrepancy”* or *”this measurement might not imply causation”*, to underscore the nuanced nature of the problem and to acknowledge the limitations inherent in the techniques we employ.
>
> We agree that an in-depth study of the factors that influence multilingual performance in LLMs is extremely important. However, given the breadth of topics covered in our paper, delving deeply into this specific aspect falls beyond the scope of our current work. Once again, we sincerely thank the reviewer for their insightful comments and will incorporate their feedback regarding the presentation of the paper.

---

### Meta-Review · Area_Chair_ZcfX · 2023-09-26

**Recommendation:** 4

**Metareview:**

The paper presents the first large scale evaluation framework for multilingual NLP tasks. Reviewers see merit in the proposal and mostly raise minor issues. Multilingual evaluation is a highly important topic in the field and the paper presents a sound account of the issue.

---

### Decision · Program_Chairs · 2023-10-07

**Decision:**

Accept-Main

**Comment:**

The paper presents the first large scale evaluation framework for multilingual NLP tasks. Reviewers see merit in the proposal and mostly raise minor issues. Multilingual evaluation is a highly important topic in the field and the paper presents a sound account of the issue.